# OTULIN maintains skin homeostasis by controlling keratinocyte death and stem cell identity

Esther Hoste [1,2,3,7,8✉], Kim Lecomte[1,2,7], Karl Annusver [4], Niels Vandamme[1,5], Jana Roels [1,5], Sophia Maschalidi [1,2], Lien Verboom [1,2], Hanna-Kaisa Vikkula[1,2], Mozes Sze[1,2], Lisette Van Hove[1,2], Kevin Verstaen[1,5], Arne Martens[1,2], Tino Hochepied[1,2], Yvan Saeys[1,5], Kodi Ravichandran[1,2,6], Maria Kasper[4] & Geert van Loo [1,2,3,8✉]

OTULIN is a deubiquitinase that specifically cleaves linear ubiquitin chains. Here we demonstrate that the ablation of *Otulin* selectively in keratinocytes causes inflammatory skin lesions that develop into verrucous carcinomas. Genetic deletion of *Tnfr1*, knockin expression of kinase-inactive *Ripk1* or keratinocyte-specific deletion of *Fadd* and *Mlkl* completely rescues mice with OTULIN deficiency from dermatitis and tumorigenesis, thereby identifying keratinocyte cell death as the driving force for inflammation. Single-cell RNA-sequencing comparing non-lesional and lesional skin reveals changes in epidermal stem cell identity in OTULIN-deficient keratinocytes prior to substantial immune cell infiltration. Keratinocytes lacking OTULIN display a type-1 interferon and IL-1β response signature, and genetic or pharmacologic inhibition of these cytokines partially inhibits skin inflammation. Finally, expression of a hypomorphic mutant *Otulin* allele, previously shown to cause OTULIN-related autoinflammatory syndrome in humans, induces a similar inflammatory phenotype, thus supporting the importance of OTULIN for restraining skin inflammation and maintaining immune homeostasis.

---

[1] VIB Center for Inflammation Research, Ghent, Belgium. [2] Department of Biomedical Molecular Biology, Ghent University, Ghent, Belgium. [3] Cancer Research Institute Ghent (CRIG), Ghent, Belgium. [4] Department of Cell and Molecular Biology, Karolinska Institutet, Stockholm, Sweden. [5] Department of Applied Mathematics, Computer Sciences and Statistics, Ghent University, Ghent, Belgium. [6] Center for Cell Clearance and Department of Microbiology, Immunology and Cancer Biology, University of Virginia, Charlottesville, Virginia, USA. [7] These authors contributed equally: Esther Hoste, Kim Lecomte. [8] These authors jointly supervised this work: Esther Hoste, Geert van Loo. ✉email: esther.hoste@irc.vib-ugent.be; geert.vanloo@irc.vib-ugent.be

The skin protects our body from external insults and against dehydration. Keratinocytes, the epithelial cells of the skin, undergo a tightly regulated differentiation program that enables the formation of a fully functional epidermal permeability barrier. Keratinocyte stem cells replace cells that have been lost through normal differentiation or programmed cell death. In recent years, a wide variety of keratinocyte stem cells have been identified that reside in different skin compartments. However, in pathophysiological conditions, such as in inflammation, wounding, or tumorigenesis, stem cells can display a tremendous plasticity and perform functions that are not part of their homeostatic repertoire[1–3]. The molecular mechanisms underlying stem cell plasticity in different pathophysiological skin states are largely unknown.

Linear ubiquitination represents a post-translational modification that is characterized by the addition of methionine (M1)-linked ubiquitin chains on protein substrates by the linear ubiquitin chain assembly complex (LUBAC)[4]. LUBAC, which consists of the proteins SHARPIN, HOIP, and HOIL-1, is the only currently known E3 ligase complex responsible for M1 ubiquitin chain formation[5–7], and mice deficient for SHARPIN or lacking HOIL-1 or HOIP in keratinocytes develop severe dermatitis[8–10]. Indeed, LUBAC-mediated linear ubiquitination controls the activation of the pro-inflammatory NF-κB pathway, but also prevents tumor necrosis factor receptor 1 (TNFR1)-mediated cell death[4,11]. OTULIN (OTU deubiquitinase with linear linkage specificity, also known as Fam105b or Gumby) is the sole deubiquitinase that exhibits a unique affinity for linear ubiquitin chains[12–14]. Patients harboring loss-of-function mutations in OTULIN develop a severe auto-inflammatory disease with skin involvement, termed ORAS (OTULIN-related auto-inflammatory syndrome, also known as otulipenia)[15,16]. Mice deficient for OTULIN or expressing a catalytically inactive OTULIN mutant die midgestation as a result of aberrant cell death mediated by TNFR1-signaling and receptor-interacting protein kinase 1 (RIPK1) kinase activity[13,17]. OTULIN directly binds to HOIP, and downregulation of OTULIN results in enhanced M1-linked ubiquitination of LUBAC and its substrates. These findings indicate that OTULIN promotes LUBAC activity by inhibiting LUBAC autoubiquitination and degradation[17–20].

Recent data point to the fact that cell death and inflammation are intricately linked, and cell death mechanisms have been shown to initiate inflammatory responses[21]. The balance between pro-inflammatory gene activation and cell death relies on signal transduction by death receptors, such as TNFR1[21,22]. The binding of TNF to TNFR1 induces the formation of the TNFR1 signaling complex, also termed complex I. Various adaptor proteins are sequentially recruited into this complex, resulting in the activation of pro-inflammatory NF-κB and MAPK signaling. Ubiquitination of distinct proteins in this complex is paramount for its assembly and downstream signaling. However, TNF can also induce inflammation by promoting cell death. In these circumstances, a different molecular complex is assembled, resulting in the formation of an apoptosis-inducing complex IIa, consisting of FADD (Fas-associated death domain) and caspase-8, or complex IIb, which relies on FADD and RIPK1 enzymatic activity, or in the formation of a necroptosis-inducing complex (termed necrosome) that depends on RIPK1 and RIPK3 kinase-activity and subsequent phosphorylation of MLKL (mixed lineage kinase domain-like)[21,22]. Genetic studies in mice have revealed that defects in proper cell death regulation may induce severe inflammatory skin phenotypes caused by keratinocyte apoptosis and necroptosis, demonstrating that keratinocyte death is a potent trigger of skin inflammation and pathology[23].

Here, we set out to investigate the importance of OTULIN-mediated linear deubiquitination for skin homeostasis by selectively deleting OTULIN in keratinocytes ($\Delta^{Ker}$OTULIN) in mice. $\Delta^{Ker}$OTULIN mice develop delineated inflammatory skin lesions from young age on that progress into verrucous carcinomas. Through genetic and pharmacological intervention studies, and by performing single-cell analysis on lesional and non-lesional skin of $\Delta^{Ker}$OTULIN mice, we identify the signaling pathways through which these lesions appear, allowing us to get new insights on the molecular events that regulate skin homeostasis and mediate skin inflammation.

## Results

**$\Delta^{Ker}$OTULIN mice exhibit inflammatory skin lesions that develop into verrucous carcinomas.** To investigate the role of OTULIN in the epidermis, $Otulin^{fl/fl}$ mice[24] were crossed to the $Keratin-14$ $Cre$ line to enable Cre-mediated recombination and OTULIN deletion selectively in keratinocytes[25]. Immunoblot analysis of primary keratinocyte cultures isolated from keratinocyte-specific OTULIN-deficient ($\Delta^{Ker}$OTULIN) mice revealed efficient deletion of OTULIN (Supplementary Fig. 1a). $\Delta^{Ker}$OTULIN mice were born with normal Mendelian segregation, but developed delineated inflamed skin lesions on back and tail skin (Fig. 1a). These skin lesions in $\Delta^{Ker}$OTULIN mice could already be observed from postnatal day (P) 6 onwards (Supplementary Fig. 1b). Skin pathology was confirmed by histology on lesional back skin from adult 7-week old $\Delta^{Ker}$OTULIN mice revealing marked epidermal hyperplasia and melanophagy (Fig. 1b). These inflammatory skin lesions progressively developed into verrucous carcinoma (Fig. 1b), defined as a well-differentiated variant of squamous cell carcinoma with minimal metastatic potential[26], and mice had to be sacrificed prior to tumor formation due to ethical concerns. Skin inflammation in $\Delta^{Ker}$OTULIN mice was also evident based on the enhanced epidermal thickness of lesional skin of these mice, while the epidermis of non-lesional skin was not thickened and was comparable to the skin of control (OTULIN$^{fl/fl}$) littermate mice (Fig. 1c). $\Delta^{Ker}$OTULIN mice also showed a loss in permeability barrier integrity in lesional skin, as assessed by transepidermal water loss (TEWL) measurements (Fig. 1d). Dermatitis in $\Delta^{Ker}$OTULIN skin was further confirmed by the marked presence of CD11b- and F4/80-positive macrophages in $\Delta^{Ker}$OTULIN skin lesions (Fig. 1e), and aberrant keratinocyte differentiation could be demonstrated in both lesional and non-lesional skin of $\Delta^{Ker}$OTULIN mice based on abnormal Keratin-6 (K6) and filaggrin staining of skin sections (Fig. 1e). $\Delta^{Ker}$OTULIN skin also exhibited substantial hypersebacea relative to the skin of control mice, as assessed by Oil Red O staining (Supplementary Fig. 1c).

Quantitative PCR analysis on epidermal tail lysates showed an increase in the expression levels of interleukin (IL)-4 and -13, which are both linked to epidermal barrier function[27], the pro-inflammatory cytokines TNF and IL-6, the chemokine MCP-1 (Monocyte Chemoattractant Protein-1, also known as CCL2), and the antimicrobial peptide S100A8 (Fig. 1f) in $\Delta^{Ker}$OTULIN skin compared to control skin, confirming the loss of barrier integrity and the inflammatory condition of $\Delta^{Ker}$OTULIN skin. $\Delta^{Ker}$OTULIN mice also exhibited enhanced circulating levels of IL-6, TNF, MCP-1, and IL-17 relative to control mice (Fig. 1g), indicating systemic inflammation, which was also apparent by the bigger size of the skin-draining lymph nodes in $\Delta^{Ker}$OTULIN mice compared to control mice (Supplementary Fig. 1d). In agreement with the inflammatory condition of $\Delta^{Ker}$OTULIN skin, immunoblotting on epidermal tail lysates revealed an enhanced NF-κB response, as evidenced by the reduced level of IκBα and increased level of phospho-IκBα in $\Delta^{Ker}$OTULIN epidermis relative to control epidermis (Fig. 1h). Also, the levels of M1-linked ubiquitin chains were markedly increased in

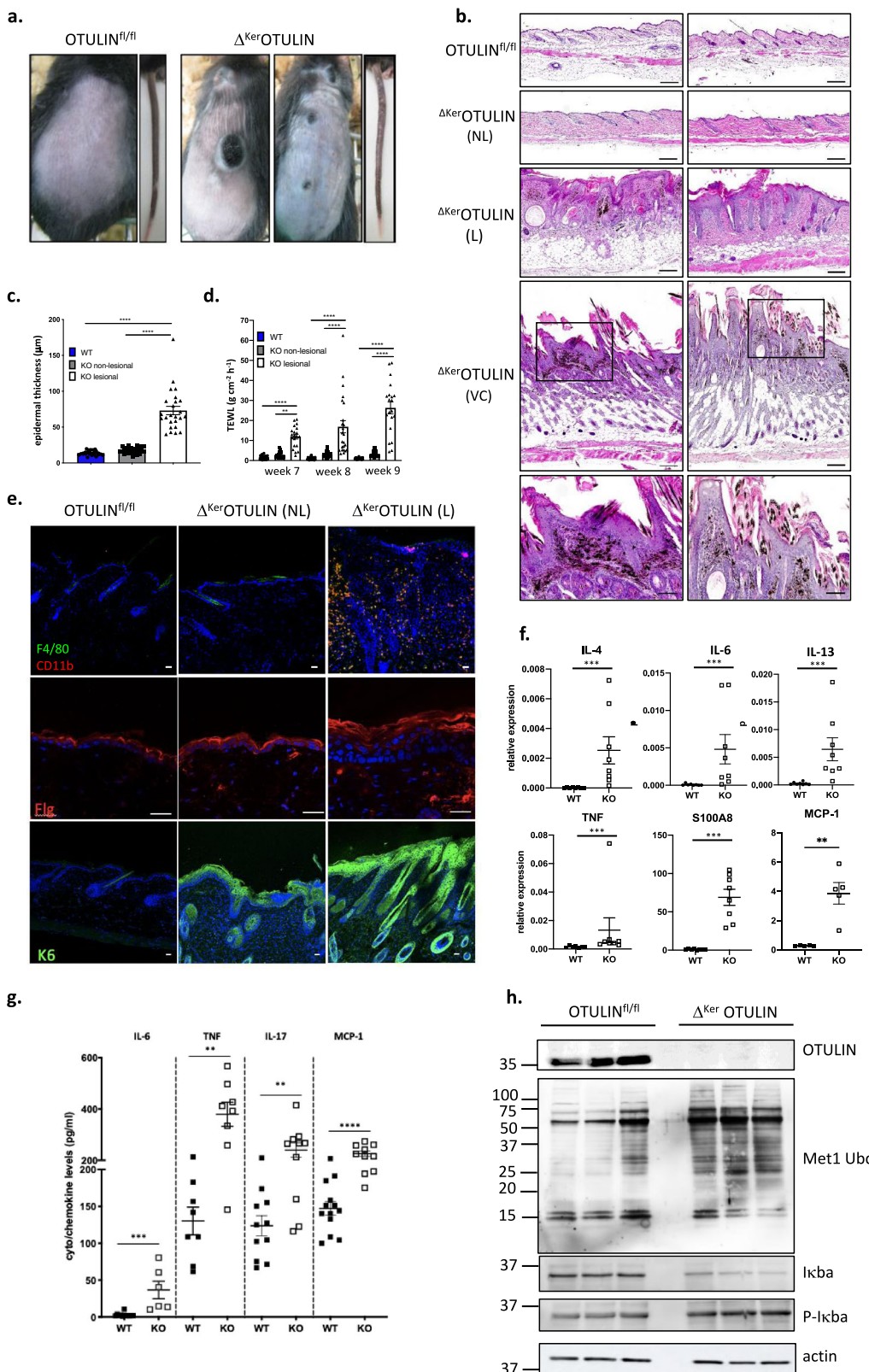

epidermal lysates from Δ^KerOTULIN mice (Fig. 1h), in agreement with the function of OTULIN as an M1 ubiquitin-specific deubiquitinase. Together, these data demonstrate the development of a strong but delineated dermatitis in mice that lack OTULIN in keratinocytes, suggesting that proper regulation of LUBAC-mediated linear ubiquitination is needed in order to maintain skin homeostasis.

**Ablation of OTULIN in keratinocytes results in enhanced epidermal stem cell proliferation and cell death**. LUBAC-mediated linear ubiquitination proved to be important for the prevention of inflammation-induced cell death in the skin[8]. OTULIN has also been recognized for its role in limiting inflammatory cell death[17,24,28,29]. Therefore, we quantified the amount of cleaved caspase-3-positive apoptotic cells in

**Fig. 1 $\Delta^{Ker}$OTULIN mice exhibit delineated inflammatory skin lesions that develop into verrucous carcinomas. a** Representative images of the back skin and tail of 7-week old OTULIN$^{fl/fl}$ and $\Delta^{Ker}$OTULIN mice. **b** Representative images of H&E-stained skin sections of 7-week old mice of the indicated genotypes. Scale bar: 100 µm. The two lower panels show verrucous carcinomas in 11-week old $\Delta^{Ker}$OTULIN mice. Boxed area depicts magnified area shown below. Scale bar for magnified areas: 200 µm. NL non-lesional; L lesional; VC verrucous carcinoma. **c** Epidermal thickness quantification at 7 weeks of age ($n = 5$ per condition; ****$p < 0.0001$, One-way ANOVA with multiple comparisons). Data represent means ± SEM. **d** Trans-epidermal water loss (TEWL) measurements at week 7, 8 and 9 of age ($n = 5$ mice per condition; **$p = 0.0019$; ****$p < 0.0001$, One-way ANOVA with multiple comparisons). Data represent means ± SEM. **e** Immunofluorescent staining of skin sections from 7-week old OTULIN$^{fl/fl}$ mice and non-lesional (NL) and lesional (L) skin of $\Delta^{Ker}$OTULIN mice with antibodies against CD11b (red) and F4/80 (green) (upper panel), filaggrin (middle panel) or keratin-6 (lower panel) and nuclear staining with Dapi. Each staining was performed on $n = 5$ mice per condition. Scale bar: 25 µm. **f** Relative mRNA expression of IL-4, IL-6, IL-13, TNF, S100A8, and MCP-1 in epidermal tail lysates of 8-weeks-old OTULIN$^{fl/fl}$ (WT; $n = 8$ and $n = 5$ for MCP-1 analysis) and $\Delta^{Ker}$OTULIN (KO; $n = 8$) mice. Data represent means ± SEM. (***$p < 0.001$; Mann−Whitney two-sided testing). **g** Levels of IL-6, TNF, IL-17, and MCP-1 in serum of 8-weeks-old OTULIN$^{fl/fl}$ (WT; $n \geq 7$) and $\Delta^{Ker}$OTULIN (KO; $n = 8$) mice (**$p < 0.01$; ***$p < 0.001$; ****$p < 0.0001$; Mann−Whitney two-sided testing was performed between relevant genotypes). **h** Western blotting on epidermal tail lysates of 7-week old OTULIN$^{fl/fl}$ ($n = 3$) and $\Delta^{Ker}$OTULIN ($n = 3$) mice using antibodies detecting OTULIN, linear ubiquitin chains (M1-Ubq), IκBα, and phospho-IκBα. Immunoblotting for phospho-IκBα and consecutively for IκBα was performed on the same blot. Molecular weight marker units are in kilodalton (kD). Anti-actin is shown as loading control. The representative image is shown for three independent experiments.

$\Delta^{Ker}$OTULIN and control skin. A significant accumulation in the number of apoptotic cells could be demonstrated in non-lesional $\Delta^{Ker}$OTULIN skin compared to control skin, which was even more pronounced in inflammatory $\Delta^{Ker}$OTULIN skin lesions (Fig. 2a, b; Supplementary Fig. 2a). Caspase-3 cleavage was confirmed by immunoblotting on epidermal tail lysates from $\Delta^{Ker}$OTULIN mice (Fig. 2c). Enhanced cell death rates in tissues are often accompanied by compensatory cell proliferation. Analysis of $\Delta^{Ker}$OTULIN skin sections also showed a marked increase in keratinocyte proliferation in both lesional and non-lesional skin regions, as evidenced by Ki67 staining (Fig. 2a, b; Supplementary Fig. 2b). In agreement, we assessed keratinocyte proliferation dynamics by pulsing $\Delta^{Ker}$OTULIN and control skin with the nucleotide analog EdU (5-ethynyl-2′-deoxyuridine) for 3 h prior to analysis. Wholemount immunofluorescence of hair follicles revealed no visible EdU uptake in control skin, while $\Delta^{Ker}$OTULIN skin showed extensive EdU uptake, indicating a strong increase in epidermal stem cell proliferation (Fig. 2d). Imaging of tail wholemounts also revealed clear abnormalities in hair follicle structures in $\Delta^{Ker}$OTULIN skin, with aberrantly shaped sebaceous glands and marked thickening of the infundibulum (Fig. 2d). Immunofluorescent staining for cleaved caspase-3 on tail wholemount sections confirmed the accumulation of caspase-3-positive dying cells over the entire length of $\Delta^{Ker}$OTULIN hair follicles, indicating that in absence of OTULIN, cell viability of hair follicle stem cell (HFSC) populations may be affected (Fig. 2e). To assess whether aberrant cell death precedes the formation of skin lesions in $\Delta^{Ker}$OTULIN mice, we next quantified the number of apoptotic cells in mice at a time-point when lesions were not yet apparent, namely at postnatal day P0.5. While epidermal thickness was not significantly altered yet in these newborn $\Delta^{Ker}$OTULIN mice, a marked increase in the number of cleaved caspase-3-positive apoptotic interfollicular epidermis (IFE) cells could already be observed in the skin of these mice compared to control mice (Fig. 2f−h).

Keratinocyte hyperproliferation in $\Delta^{Ker}$OTULIN mice is indicative for an increase in stem cell proliferative capacity, which is crucial for regenerative responses in the skin[1]. Indeed, full-thickness skin wounding in $\Delta^{Ker}$OTULIN and control mice revealed a marked accelerated wound closure response in the initial phases of wound repair (day 2 and day 4 post-wounding) in $\Delta^{Ker}$OTULIN mice. However, when wounds enter the remodeling stage of repair (day 8 post-wounding), wound closure slowed down significantly in $\Delta^{Ker}$OTULIN skin compared to control skin (Supplementary Fig. 2c−e). Intriguingly, $\Delta^{Ker}$OTULIN skin developed cysts and tumor-like lesions at sites of wounding when re-epithelialization was complete

(Supplementary Fig. 2e), confirming the enhanced sensitivity of these mice to skin tumorigenesis.

**TNFR1-mediated cell death drives inflammation in $\Delta^{Ker}$OTULIN mice.** The cutaneous inflammation in Sharpin$^{cpdm/cpdm}$ mice does not develop in the absence of TNFR1[5,30,31]. However, the lethal dermatitis present in keratinocyte-specific HOIL-1 or HOIP deficient mice is only partially mediated by TNFR1[8]. Therefore, we tested whether genetic ablation of TNFR1 also results in an amelioration of the inflammatory phenotype observed in $\Delta^{Ker}$OTULIN skin. Crossing $\Delta^{Ker}$OTULIN mice onto a TNFR1-deficient background completely prevented dermatitis (Fig. 3a), even at old age (Supplementary Fig. 3a), and $\Delta^{Ker}$OTULIN-TNFR1$^{-/-}$ mice showed significantly reduced IL6, TNF, and IL17 levels in their serum (Fig. 3b). Moreover, deletion of one functional TNFR1 allele partially protected $\Delta^{Ker}$OTULIN skin against the formation of skin lesions and inflammatory cytokine production (Fig. 3b). In agreement, the epidermis of $\Delta^{Ker}$OTULIN-TNFR1$^{-/-}$ skin was not thickened and was comparable to the skin of control (OTULIN$^{fl/fl}$) littermate mice (Fig. 3c).

Since RIPK1 kinase activity regulates cell death in the TNFR1-complex II, we next evaluated the contribution of RIPK1 kinase-dependent cell death to the inflammatory skin phenotype of $\Delta^{Ker}$OTULIN mice. For this, we crossed $\Delta^{Ker}$OTULIN mice onto a RIPK1 kinase-inactive genetic ($Ripk1^{D138N/D138N}$) background. $\Delta^{Ker}$OTULIN-RIPK1$^{D138N/D138N}$ mice were also completely protected from dermatitis development (Fig. 3a), even at old age (40 weeks and older) (Supplementary Fig. 3b), and showed normal serum levels of inflammatory cytokines and chemokines (Fig. 3b), and normal thickness of the epidermis (Fig. 3c). RIPK1 kinase activity can induce both FADD-dependent apoptosis and RIPK3/MLKL-dependent necroptosis[32]. Hence, we next crossed $\Delta^{Ker}$OTULIN mice with mice that have a floxed *Fadd* and *Mlkl* allele, generating mice that lack OTULIN, FADD, and MLKL specifically in keratinocytes. These $\Delta^{Ker}$OTULIN/FADD/MLKL mice were completely lesion-free (Fig. 3a and Supplementary Fig. 3c), exhibited normal circulating cytokine levels (Fig. 3b), and normal thickness of the epidermis (Fig. 3c), proving that cell death is the driving force of inflammation in $\Delta^{Ker}$OTULIN mice. $\Delta^{Ker}$OTULIN mice only deficient for MLKL in keratinocytes were partially protected from dermatitis. The tail phenotype was completely rescued in $\Delta^{Ker}$OTULIN/MLKL mice, however some mice still developed lesions on the back skin (Supplementary Fig. 3d). Finally, we observed complete protection from dermatitis when $\Delta^{Ker}$OTULIN mice were crossed onto a MyD88-deficient genetic background, suggesting that microbial

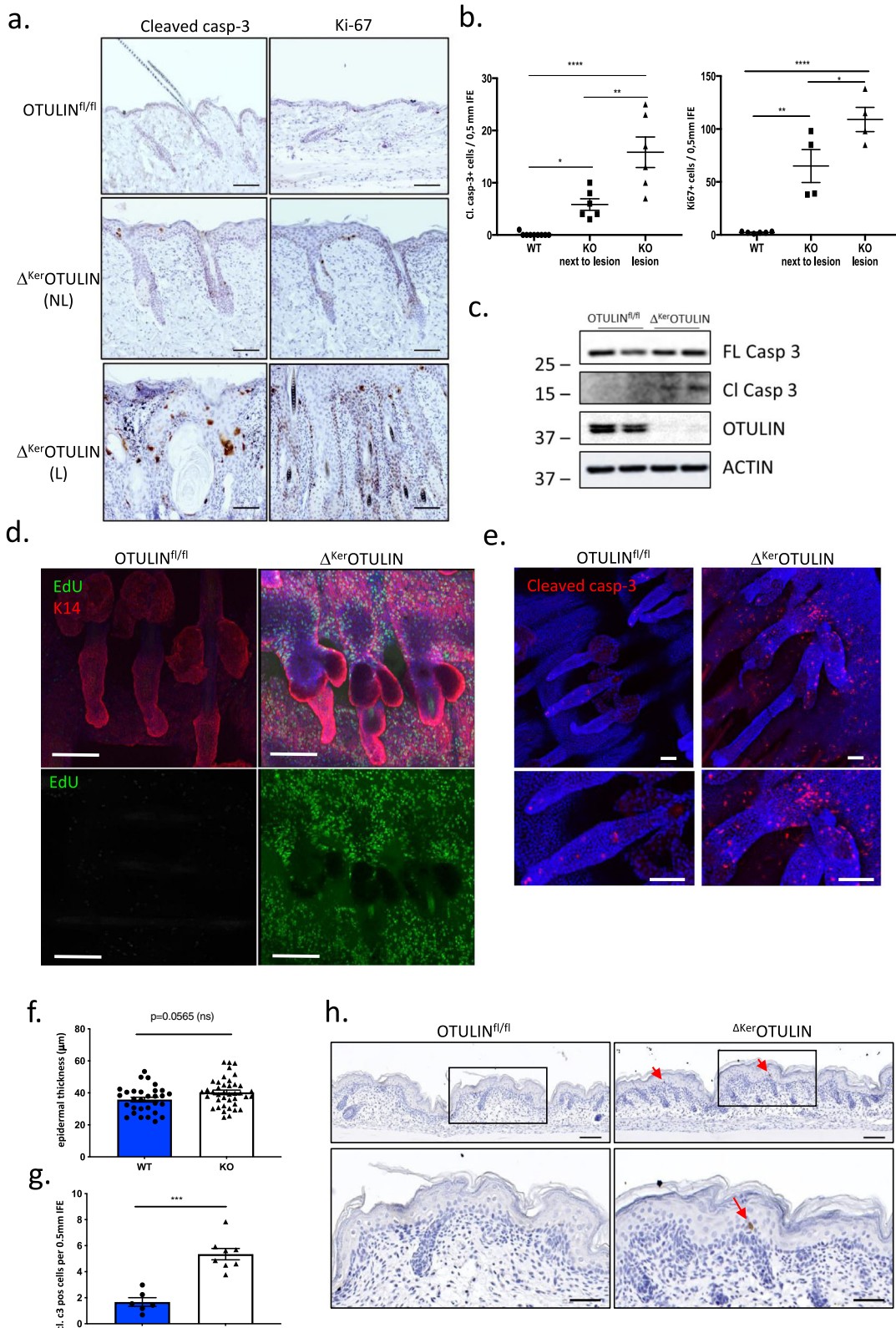

components could be involved in driving dermatitis in OTULIN-deficient skin (Fig. 3a).

Primary mouse keratinocytes (PMKs) isolated from *Shar pin^{cpdm/cpdm}* mice are highly sensitive to cell death induced by TNF stimulation[6,33], and HOIP-deficient cells are less viable even in the absence of exogenous stimuli[8]. OTULIN-deficient PMKs, however, are equally resistant to TNF-induced cell death as control

PMKs (Fig. 3d). However, when PMKs were primed with type II IFN (IFNγ) and subsequently stimulated with TNF, a significant larger amount of OTULIN-deficient keratinocytes were shown to die compared to control PMKs, as measured by uptake of a cell impermeable dye over a 24 h time course (Fig. 3d, e). In contrast, no significant differences in NF-κB and p38 MAPK signaling or production of cytokines or chemokines could be observed between

**Fig. 2 Epidermis of $\Delta^{Ker}$OTULIN mice exhibits hyperproliferation and enhanced apoptosis. a** Skin sections of 7-week old OTULIN$^{fl/fl}$ and $\Delta^{Ker}$OTULIN mice stained with antibodies against cleaved caspase-3 and Ki-67 to assess apoptosis and proliferation, respectively. NL non lesional; L lesional. Scale bar: 200 μm. **b** Quantification of the number of interfollicular epidermis (IFE) cells from 7-week old OTULIN$^{fl/fl}$ (WT) and $\Delta^{Ker}$OTULIN (KO) mice that stain positive for cleaved caspase-3 and Ki-67 ($n = 9$ WT mice; $n = 6$ and 4 KO non-lesional mice per condition; $n = 6$ and 4 KO lesional mice; One-way ANOVA; *$p < 0.5$; **$p < 0.01$; ****$p < 0.0001$). Data represent means ± SEM. **c** Western blot analysis for expression of full-length (FL) and cleaved (Cl) caspase-3 in epidermal tail lysates from OTULIN$^{fl/fl}$ and $\Delta^{Ker}$OTULIN mice. Anti-actin immunoblotting was used as a loading control. Molecular weight marker units are in kilodalton (kD). This experiment was repeated three times independently with similar results. **d** EdU (green) retaining cells in tail wholemount sections of 7-week old OTULIN$^{fl/fl}$ and $\Delta^{Ker}$OTULIN mice after 3 h chase. Wholemounts were stained for Keratin-14 (K14; red) and counterstained with Dapi (blue). Scale bars: 100 μm. Lower panels show EdU staining only. This experiment was repeated three times independently with similar results. **e** Tail wholemount sections of 7-week old OTULIN$^{fl/fl}$ and $\Delta^{Ker}$OTULIN mice stained for cleaved caspase-3 (red) and counterstained with Dapi. Lower panels depict magnified views. Eight mice per genotype were analyzed. Scale bars: 50 μm. **f, g** Quantification of epidermal thickness (**f**) and the number of cleaved caspase-3-positive cells (**g**) in the IFE of skin sections of P0.5 control and $\Delta^{Ker}$OTULIN pups (WT: $n = 3$; KO: $n = 4$; Mann−Whitney two-sided test; ***$p = 0.0007$). Data represent means ± SEM. **h** Representative images of cleaved capase-3-stained skin sections of OTULIN$^{fl/fl}$ and $\Delta^{Ker}$OTULIN pups. Eight mice per genotype were analyzed. Arrows indicate apoptotic IFE cells. Lower panels depict magnified views of the boxed areas. Scale bars: 100 μm.

control and $\Delta^{Ker}$OTULIN PMKs after stimulation with TNF (Fig. 3f and Supplementary Fig. 3e). It should be noted that the residual OTULIN band observed in $\Delta^{Ker}$OTULIN PMKs might originate from feeder cells that can still be present in PMK cultures. A pronounced reduction could be observed in the expression of SHARPIN, HOIL-1, and HOIP in $\Delta^{Ker}$OTULIN PMKs (Fig. 3f), in agreement with previous studies that have shown reduced expression of LUBAC components in OTULIN-deficient cells and tissues[15,17,24,28,29], and consistent with the concept that OTULIN maintains LUBAC function by suppressing its auto-ubiquitination and degradation[17]. Indeed, pretreatment of PMKs with the proteasome inhibitor MG132 could restore SHARPIN and HOIP levels in $\Delta^{Ker}$OTULIN cultures (Supplementary Fig. 3f). Finally, analysis of linear ubiquitination by specific pulldown of ubiquitin-binding domain-containing proteins using recombinant GST-UBAN (Ub-binding domain in ABIN proteins and NEMO), demonstrated a strong increase in M1-ubiquitination in primary OTULIN-deficient keratinocyte cultures after stimulation with TNF, compared to cultures from control mice (Fig. 3g), confirming the importance of OTULIN in restricting M1 ubiquitination in keratinocytes. Immunoblotting for RIPK1 in immunoprecipitation lysates of PMKs showed a decreased expression of RIPK1 in $\Delta^{Ker}$OTULIN PMKs (Fig. 3g), although this altered RIPK1 expression was not observed in epidermal tails lysates isolated from $\Delta^{Ker}$OTULIN mice (Supplementary Fig. 3g).

In conclusion, we could demonstrate that dermatitis and tumor development in $\Delta^{Ker}$OTULIN mice depends on the cytotoxic activity of TNF driving FADD- and RIPK1 kinase-dependent death of keratinocytes. The complete rescue from dermatitis upon genetic deletion of both FADD and MLKL proves that keratinocyte cell death is the driving force of the skin inflammation and tumorigenesis in $\Delta^{Ker}$OTULIN mice.

**OTULIN-deficiency in keratinocytes perturbs stem cell lineage and induces cutaneous infiltration of innate immune cells.** To better characterize the inflammatory phenotype of $\Delta^{Ker}$OTULIN mice and to gain insights into the cellular differences between lesional and non-lesional $\Delta^{Ker}$OTULIN skin, we next performed single-cell RNA-sequencing (scRNAseq) on live cells sorted from control wild-type (WT, OTULIN$^{fl/fl}$; $n = 1$) skin and lesional (L; $n = 3$) and non-lesional (NL; $n = 2$) $\Delta^{Ker}$OTULIN skin. Following pre-processing of the data according to the Marioni pipeline, poor quality cells were excluded[34]. Firstly, unsupervised global clustering into populations was performed with affinity propagation according to the expression of high variance genes (Fig. 4a). The different cell populations that were delineated by unbiased clustering were annotated according to the expression of cell markers adapted from Joost et al.[35] (Supplementary Fig. 4a).

Next, we determined which cells originated from the three different conditions. This analysis revealed a marked clustering of the control (WT) cells within the different cell populations, which was opposite to the distribution of the lesional cells (L) within the clusters. This was remarkably clear in the keratinocyte and fibroblast clusters (Fig. 4b). Interestingly, $\Delta^{Ker}$OTULIN non-lesional (NL) cells were distributed over the entire keratinocyte and fibroblast cluster (Fig. 4b, middle panel). These cell-types are undergoing major changes in overall gene expression in non-lesional $\Delta^{Ker}$OTULIN skin, resulting in the presence of cells with expression profiles that are highly similar to WT cells, alongside cells that are highly similar to lesional cells and cells that are clearly transitioning in between these two ends of the expression profile spectrum.

scRNAseq confirmed a strong infiltration of innate immune cells in lesional $\Delta^{Ker}$OTULIN skin, while in non-lesional $\Delta^{Ker}$OTULIN skin only a slight increase in these immune cells could be observed relative to the situation in WT skin (Fig. 4b, c). Subclustering and annotation of this immune cell population revealed that this cluster harbors mainly macrophages and dendritic cells (Fig. 4c). The gradual infiltration of immune cells in $\Delta^{Ker}$OTULIN skin demonstrates the progressive nature of the inflammatory phenotype that develops in these mice. The higher abundance of innate immune cells in lesional $\Delta^{Ker}$OTULIN skin was confirmed by flow cytometry, showing a significant increase in the total number of CD45+ immune cells in lesional skin versus non-lesional $\Delta^{Ker}$OTULIN or control skin (Fig. 4d). A marked infiltration of F4/80-positive macrophages, cDC1, cDC2, eosinophils, and Langerhans cells was observed in lesional $\Delta^{Ker}$OTULIN skin relative to non-lesional $\Delta^{Ker}$OTULIN or control skin in both flow cytometry and scRNAseq (Fig. 4d, e; Supplementary Fig. 4b). Interestingly, the most substantial difference that could be observed in lesional $\Delta^{Ker}$OTULIN versus non-lesional skin according to flow cytometric analyses was an increase in the number of inflammatory macrophages (CD45+ CD11b+ F4/80+ cells) (Fig. 4d). scRNAseq analysis also revealed extensive changes in the T-cell population, where a substantial infiltration of regulatory T-cells (Tregs) occurred in both lesional and non-lesional $\Delta^{Ker}$OTULIN skin (Supplementary Fig. 4c−e), which was confirmed by flow cytometric quantification of the number of FoxP3+ T-cells (Supplementary Fig. 4f).

Our scRNAseq data also showed that several keratinocyte populations marked by HFSC markers, such as Lgr5+, Lrig1+, and Sox9+ HFSCs, show a gradual expansion in both lesional and non-lesional skin, while others, such as CD34+ keratinocytes, gradually decrease in frequency in non-lesional and lesional $\Delta^{Ker}$OTULIN skin (Fig. 4f and Supplementary Fig. 4g). These data indicate that stem cells display a high degree of plasticity in

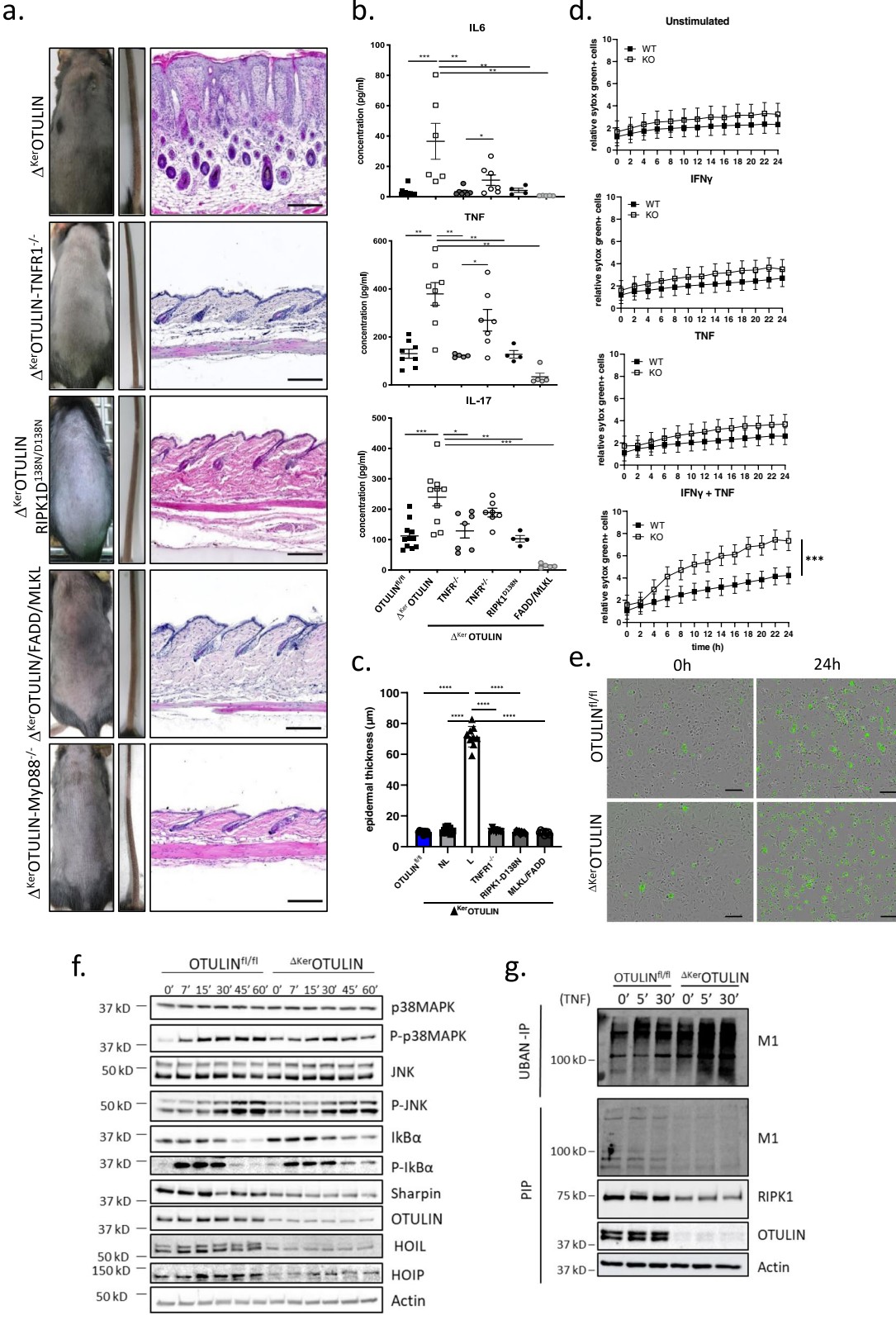

the inflammatory skin condition in $\Delta^{Ker}$OTULIN skin, suggesting that defects in proper regulation of linear ubiquitination are important for stem cell lineage in the skin. These stem cell changes are already initiated prior to the massive infiltration of immune cells into the skin, as stem cell populations exhibit transcriptional changes in non-lesional $\Delta^{Ker}$OTULIN skin that does not harbor significantly more immune cells than control skin (Fig. 4e, f).

**Type-1 interferons contribute to dermatitis in $\Delta^{Ker}$OTULIN mice.** Recent studies identified an important role for type-1

**Fig. 3 Dermatitis in $\Delta^{\text{Ker}}$OTULIN mice depends on TNFR1-signaling and on FADD/MLKL- and RIPK1 kinase-dependent cell death. a** Representative images of the back skin and tail of 7-week old mice of the indicated genotype. The right panel shows H&E-stained skin sections of the respective mice. Scale bar: 200 μm. **b** Levels of IL-6, TNF and IL-17 in serum of 8-weeks-old OTULIN$^{\text{fl/fl}}$ (WT; $n = 9$ or 11), $\Delta^{\text{Ker}}$OTULIN (KO; $n = 8$), $\Delta^{\text{Ker}}$OTULIN-TNFR1$^{-/-}$ ($n = 7$), $\Delta^{\text{Ker}}$OTULIN TNFR1$^{+/-}$ ($n = 7$), $\Delta^{\text{Ker}}$OTULIN-RIPK1$^{\text{D138N/D138N}}$ ($n = 4$), and $\Delta^{\text{Ker}}$OTULIN/FADD/MLKL ($n = 5$) mice (Mann−Whitney two-sided test; *$p < 0.5$; **$p < 0.01$; ***$p < 0.001$; ****$p < 0.0001$). Data represent means ± SEM. **c** Epidermal thickness quantification at 7−11 weeks of age. NL non-lesional; L lesional ($n = 10$ per condition; ****$p < 0.0001$, One-way ANOVA with multiple comparisons). Data represent means ± SEM. **d** Primary keratinocyte cultures ($n = 3$ biological replicates per condition) were treated with 20 ng/ml mTNF with or without priming with IFN-γ (10 ng/ml) 8 h prior to TNF stimulation. Viability was assessed by Sytox Green uptake. Representative graphs for three independent experiments. (Residual maximum likelihood (REML); ***$p = 0.002$). Data represent means ± SEM. **e** Incucyte images depicting Sytox Green uptake by dead keratinocytes at 0 and 24 h post IFN-γ and TNF treatment. This experiment was repeated three times independently with similar results. **f** Western blot analysis on lysates from primary keratinocyte cultures isolated from OTULIN$^{\text{fl/fl}}$ and $\Delta^{\text{Ker}}$OTULIN mice that were treated with TNF for the indicated time-points. The representative figure for four independent experiments. **g** Ubiquitin pulldown by UBAN-IP on lysates from PMK cultures isolated from OTULIN$^{\text{fl/fl}}$ and $\Delta^{\text{Ker}}$OTULIN mice treated with TNF for the indicated time-points followed by immunoblotting for M1 Ubq chains. Pre-immunoprecipitation lysate (PIP) was immunoblotted for M1 Ubq (M1), RIPK1, and OTULIN. α-Actin is shown as loading control. This experiment was repeated three times independently with similar results.

interferons (IFNs) in the process of inflammation[4,17,24]. Expression analysis of IFN-stimulated genes (ISGs) using our scRNAseq dataset indeed revealed an increased expression of ISGs in OTULIN-deficient keratinocytes. Remarkably, multiple ISGs such as *Irf3*, *Irf9*, and *USP18* were already upregulated in non-lesional $\Delta^{\text{Ker}}$OTULIN skin, implicating that IFN signaling is an early event in the generation of dermatitic lesions (Fig. 5a). Q-PCR analysis of $\Delta^{\text{Ker}}$OTULIN and control epidermal tail lysates confirmed the upregulation of ISGs and type-1 IFNs in keratinocytes in the absence of OTULIN (Fig. 5b).

To further investigate whether type-1 IFNs are crucial in driving the skin inflammation in $\Delta^{\text{Ker}}$OTULIN mice, these mice were crossed to *Ifnar1* (Interferon-α receptor 1)-deficient mice. $\Delta^{\text{Ker}}$OTULIN-IFNAR1$^{-/-}$ mice showed a rescue of the skin phenotype with lower lesion incidence and skin lesions developing later in life compared to $\Delta^{\text{Ker}}$OTULIN mice (Fig. 5c, d), even at old age (>40 weeks of age) (Supplementary Fig. 3h). In agreement, serum levels of IL-6, TNF, IL-17, and MCP-1 were reduced to baseline in many $\Delta^{\text{Ker}}$OTULIN IFNAR1$^{-/-}$ mice (Fig. 5e). However, while some $\Delta^{\text{Ker}}$OTULIN IFNAR1$^{-/-}$ mice were completely protected from dermatitis, others still developed skin inflammation over time (Fig. 5d−f). Finally, immunostaining for the type-1 interferon IFN-β revealed marked expression of this cytokine in lesional $\Delta^{\text{Ker}}$OTULIN skin (Fig. 5g). Collectively, these data demonstrate that the production of type-1 IFNs critically contributes to the formation of inflammatory skin lesions in $\Delta^{\text{Ker}}$OTULIN mice.

**Dermatitis in $\Delta^{\text{Ker}}$OTULIN mice is mediated by Interleukin-1β released by innate immune cells.** We next set out to identify the signals driving the response of OTULIN-deficient keratinocytes to inflammation. Therefore, we made use of the NicheNet algorithm designed to infer ligand-receptor links between interacting cells by combining transcriptome data of interacting cells with existing knowledge on gene regulatory networks[36]. NicheNet analysis was applied to predict the ligands that are produced by innate immune cells and bind to receptors on keratinocytes, causing the changes in keratinocyte gene expression profiles (Fig. 6a). One of the top predicted ligands which we identified by NicheNet to be produced by infiltrating innate immune cells in lesional skin of $\Delta^{\text{Ker}}$-OTULIN mice and modulating gene expression in keratinocytes, was the cytokine IL-1β (Fig. 6a). This cytokine was also identified by NicheNet as a ligand with putative regulatory potential in keratinocytes when comparing non-lesional skin to control skin (Supplementary Fig. 5a), indicating that IL-1β could be an early mediator of the aberrant keratinocyte behavior in $\Delta^{\text{Ker}}$OTULIN skin. We next assessed the expression profile of IL-1β and the IL-1 family members IL-1α and IL-18 in our scRNAseq dataset and could observe that IL-1β was indeed strongly produced by

immune cells infiltrating lesional skin of $\Delta^{\text{Ker}}$OTULIN mice (Fig. 6b and Supplementary Fig. 5b). This expression profiling also revealed that macrophages represent the predominant IL-1β-producing cell population (Fig. 6b). Interestingly, next to IL-1β and IL-18, also other genes involved in inflammasome activation and IL-1β production, including caspase-1, ASC, and Nlrp3, were upregulated in keratinocytes of lesional skin of $\Delta^{\text{Ker}}$OTULIN mice relative to keratinocytes from non-lesional $\Delta^{\text{Ker}}$OTULIN and wild-type mice (Fig. 6c). To assess the functional importance of IL-1β in the pathology of $\Delta^{\text{Ker}}$OTULIN mice, we next treated $\Delta^{\text{Ker}}$OTULIN mice with Anakinra, a recombinant version of the human interleukin-1 receptor (IL1R), blocking the binding of IL-1α and IL-1β to the IL1R[37]. Daily intraperitoneal injections of $\Delta^{\text{Ker}}$OTULIN mice with Anakinra starting from P18 onwards could suppress dermatitis development in the back skin and tail of these mice, confirming an important contribution of IL-1β in the development of skin lesions (Fig. 6d and Supplementary Fig. 6). The therapeutic potency of Anakinra treatment to ameliorate dermatitis in $\Delta^{\text{Ker}}$OTULIN mice was also evidenced by a marked restoration of the epidermal permeability function in the lesional skin of these mice (Fig. 6e). These data demonstrate that IL-1β production by infiltrating immune cells contributes to the inflammatory skin phenotype in $\Delta^{\text{Ker}}$OTULIN mice.

Next to the cytokine IL-1β, we also identified the chemokine MCP-1 by NicheNet as a ligand that could mediate transcriptional changes in keratinocytes when comparing non-lesional skin to control skin (Supplementary Fig. 5a). MCP-1 is a potent chemokine attracting macrophages, therefore we investigated whether blocking MCP-1 could suppress dermatitis in $\Delta^{\text{Ker}}$O-TULIN mice. Indeed, intraperitoneal injections of $\Delta^{\text{Ker}}$OTULIN mice with a neutralizing α-MCP-1 antibody could ameliorate but not fully suppress the dermatitis in back skin, and completely rescue the inflammatory phenotype in $\Delta^{\text{Ker}}$OTULIN tail skin (Fig. 6f, g and Supplementary Fig. 6).

**Knockin of a human mutation in the murine *Otulin* gene phenocopies OTULIN-deficiency.** Homozygous hypomorphic mutations in the human *OTULIN* gene, affecting the deubiquitinase activity of the protein, have been shown to underlie the development of a severe life-threatening autoinflammatory syndrome, called ORAS[15,16]. ORAS patients develop neonatal-onset fever, swollen joints, and diarrhea, but also dermatitis and panniculitis[15,16,38]. The best characterized homozygous missense mutation L272P (c.815T > C;p Leu272Pro) was shown to result in reduced OTULIN stability and activity towards M1 linked ubiquitin, and patients' fibroblasts and peripheral blood mononuclear cells showed evidence for increased NF-κB signaling and production of inflammatory cytokines[15,16].

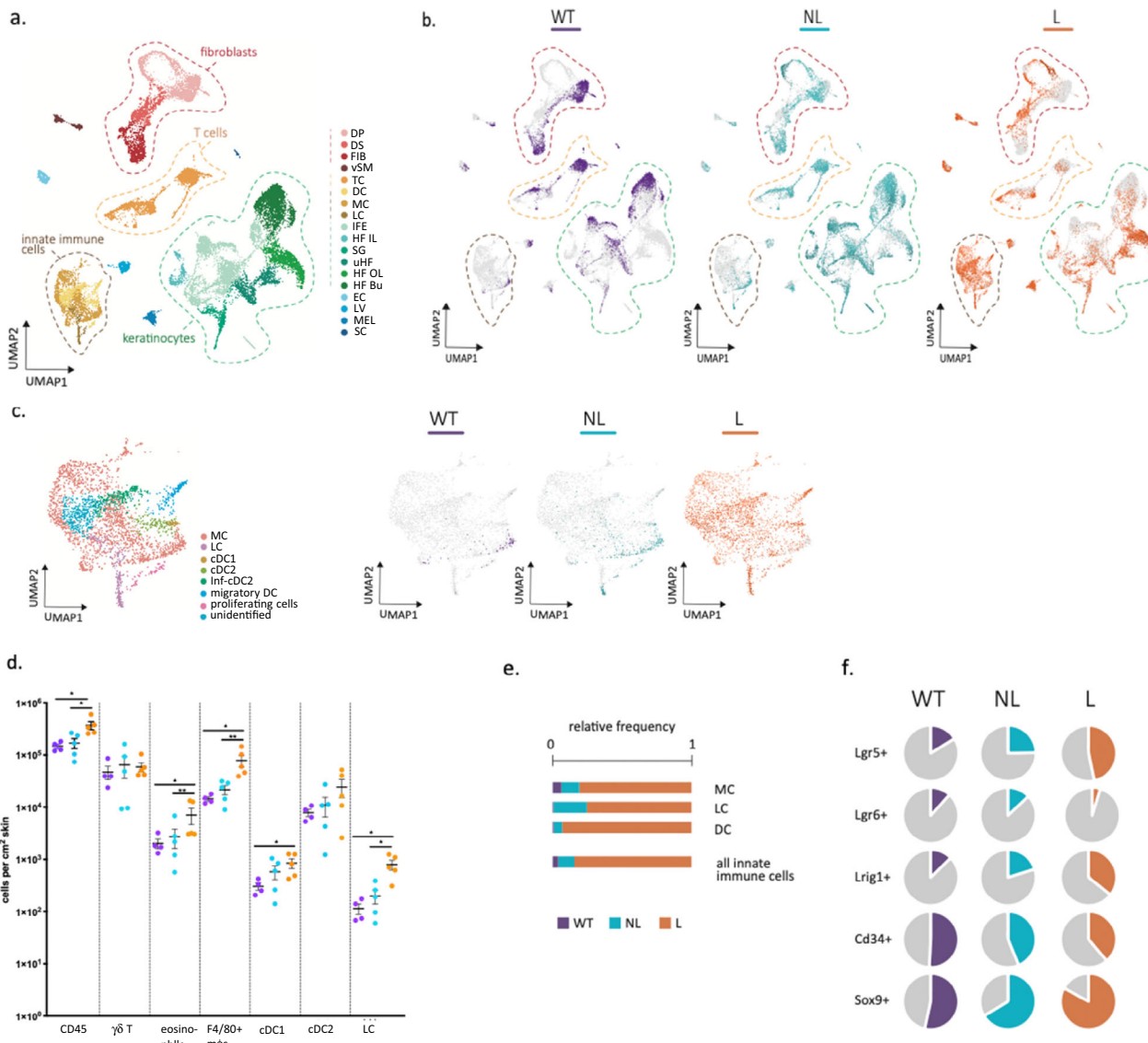

**Fig. 4 Single-cell RNA-sequencing reveals marked infiltration of innate immune cells and changes in hair follicle stem cell lineage in Δ$^{Ker}$OTULIN mice. a** Annotated UMAP clustering of live skin cells isolated from control OTULIN$^{fl/fl}$ mice (WT) and from non-lesional (NL) and lesional (L) total skin of Δ$^{Ker}$OTULIN mice (DP dermal papilla, DS dermal sheath, FIB fibroblasts, vSM vascular smooth muscle, TC T-cells, DC dendritic cells, MC macrophages, LH Langerhans cells, IFE interfollicular epidermis (including infundibulum), HF IL hair follicle anagen inner layer cells, SG sebaceous gland, uHF upper hair follicle, HF OL hair follicle anagen outer layer cells, HF Bu hair follicle bulge, EC endothelial cells, LV lymph vessel, MEL melanocytes, SC Schwann cells). **b** Distribution of WT, NL, and L cells within the cell clusters. **c** UMAP plot of annotated innate immune cell subcluster, showing assigned clusters and distribution of WT, NL, and L cells within this cluster. **d** Flow cytometric analysis of immune cell composition in control OTULIN$^{fl/fl}$ skin (WT, $n = 4$ mice) and non-lesional (NL) and lesional (L) skin of Δ$^{Ker}$OTULIN mice ($n = 5$ mice per condition). The absolute number of immune cells per cm$^2$ skin is plotted, data represent means ± SEM. (Mann−Whitney two-sided test; *$p < 0.5$; **$p < 0.01$; ***$p < 0.001$; ****$p < 0.0001$). **e** Relative normalized cell frequencies of the indicated immune cell-types in the different conditions (LH Langerhans cells, MC macrophages, DC dendritic cells). **f** percentage of keratinocytes expressing the indicated HFSC markers within the total keratinocyte population of WT, NL and L skin.

To experimentally assess the importance of the L272P mutation, we generated a novel knockin transgenic mouse line expressing the OTULIN$^{L272P}$ mutation (amino acid L272 being conserved in mouse) through CRISPR/Cas gene-editing technology. The desired point mutation was verified by PCR amplification and sequencing of the DNA sequence around the target sites (Fig. 7a). Heterozygous OTULIN$^{L272P/+}$ knockin mice were crossed to homozygosity, but no homozygous OTULIN$^{L272P/L272P}$ mice were born (Table 1), confirming the lethal phenotype caused by the loss of the deubiquitinase function in these mice, in agreement with what has been shown before in OTULIN knockout mice[15,24], in knockin mice that

express catalytically inactive (C129A) OTULIN[17], and in *gumby* mice that have a point mutation (W96R or D336E) in *Otulin* that abolishes its ability to bind to ubiquitin[13]. However, the OTULIN$^{L272P/L272P}$ lethality could be rescued when OTULIN$^{L272P}$ mice were crossed into a caspase-8 and RIPK3 deficient background (Table 1 and Fig. 7b, c). OTULIN$^{L272P/L272P}$casp8$^{-/-}$RIPK3$^{-/-}$ mice were born in Mendelian numbers and developed normally without any sign of inflammation (Table 1, Fig. 7b, c) with the exception of the lymphoproliferative syndrome that develops in Casp8$^{-/-}$RIPK3$^{-/-}$ mice, as previously shown[39,40]. This rescue implies that aberrant cell death triggers lethality in OTULIN$^{L272P/L272P}$ mice.

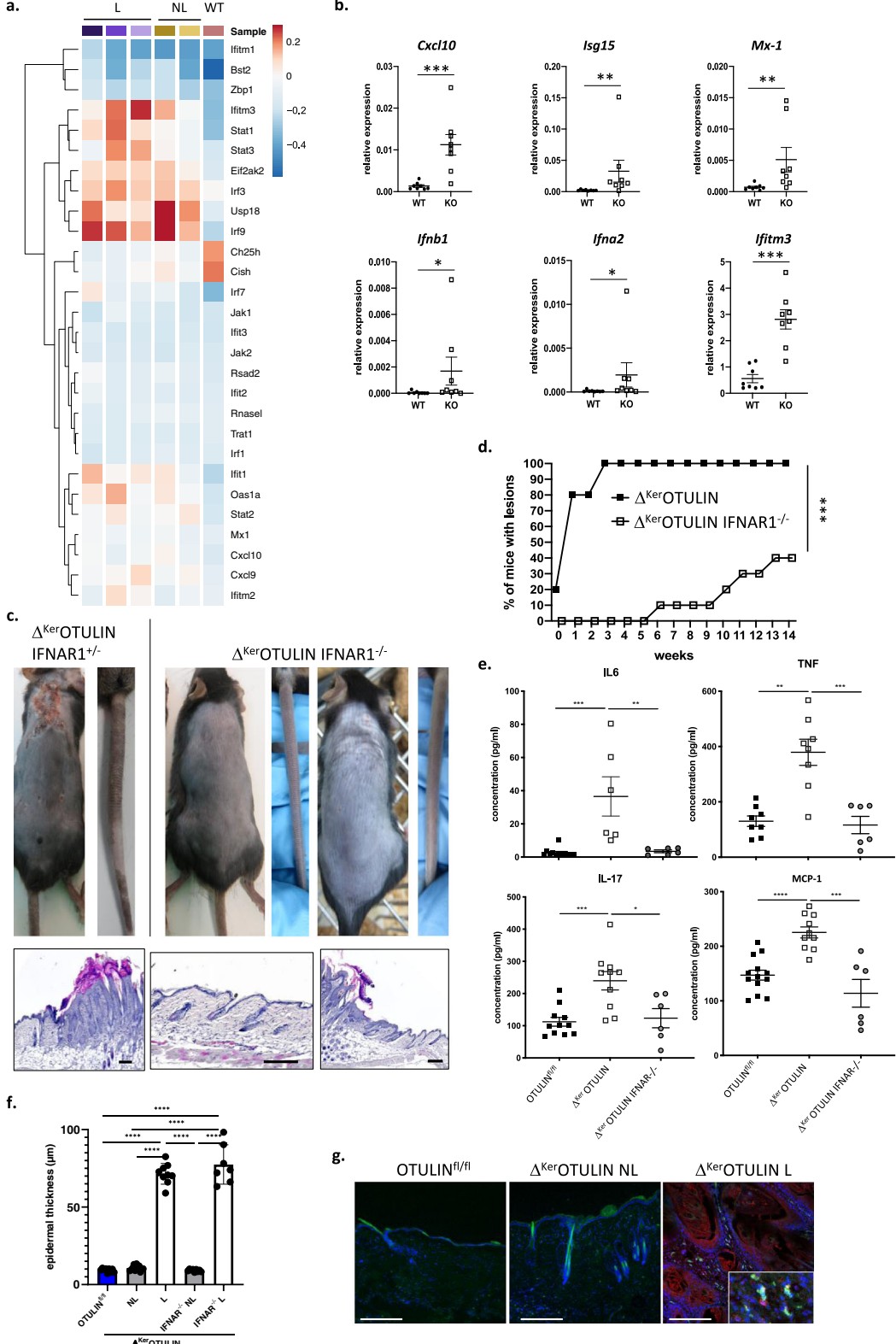

We next crossed OTULIN$^{L272P/+}$ knockin mice with heterozygous keratinocyte-specific OTULIN deficient mice, generating OTULIN$^{L272P/\Delta Ker}$ mice, having one L272P knockin allele in all cells and tissues, and one OTULIN knockout allele specifically in keratinocytes. These mice develop skin lesions and verrucous carcinomas on their back skin in a similar manner and timeframe as $\Delta^{Ker}$OTULIN mice (Fig. 7b, c). Immunoblotting on PMKs isolated from OTULIN$^{L272P/\Delta Ker}$ mice confirmed the reduced stability of OTULIN, SHARPIN, and HOIP (Fig. 7d).

Together, these findings demonstrate that expression of a human-relevant OTULIN mutation in mice induces a similar inflammatory skin phenotype as observed in keratinocyte-specific OTULIN knockout mice, confirming that proper regulation of linear protein ubiquitination is crucial for mammalian skin homeostasis.

**Fig. 5 Dermatitis in $\Delta^{\text{Ker}}$OTULIN mice is partially mediated by type-1 interferon signaling. a** Heatmap showing average expression levels of the indicated interferon-response genes in keratinocytes of the different conditions as observed in single-cell RNA-seq data. **b** Relative mRNA expression of *Cxcl-10*, *Isg15*, *Mx-1*, *Ifnb1*, *Ifna2*, and *Ifitm3* in epidermal tail lysates of 8-weeks-old OTULIN$^{fl/fl}$ (WT; $n = 8$) and $\Delta^{\text{Ker}}$OTULIN (KO; $n = 8$) mice. Data represent means ± SEM. (*$p < 0.05$; **$p < 0.01$; ***$p < 0.001$; Mann−Whitney two-sided test). **c** Representative images of the back skin and tail (upper panel), and H&E-stained skin sections (lower panel) of $\Delta^{\text{Ker}}$OTULIN-IFNAR1$^{+/-}$ and $\Delta^{\text{Ker}}$OTULIN-IFNAR1$^{-/-}$ mice. Scale bars: 200 μm. **d** Incidence plot showing the percentage of $\Delta^{\text{Ker}}$OTULIN ($n = 20$) and $\Delta^{\text{Ker}}$OTULIN-IFNAR1$^{-/-}$ ($n = 20$) mice that exhibit lesions on their skin (***$p = 0.0009$, log-rank testing). **e** Levels of IL-6, TNF, IL-17 and MCP-1 in serum of 8-weeks-old OTULIN$^{fl/fl}$ ($n = 11$), $\Delta^{\text{Ker}}$OTULIN ($n = 6$), and $\Delta^{\text{Ker}}$OTULIN IFNAR1$^{-/-}$ ($n = 6$) mice (Mann−Whitney two-sided test; *$p < 0.5$; **$p < 0.01$; ***$p < 0.001$; ****$p < 0.0001$). Data represent means ± SEM. **f** Epidermal thickness quantification of skin of 7−11 weeks old mice. NL non-lesional; L lesional ($n = 8$ or 10 per condition; ****$p < 0.0001$, One-way ANOVA with multiple comparisons). Data represent means ± SEM. **g** Immunofluorescent staining for CD45 (green) and IFN-β (red) on skin sections from OTULIN$^{fl/fl}$ mice and non-lesional (NL) and lesional (L) skin of $\Delta^{\text{Ker}}$OTULIN mice. Sections are counterstained with Dapi (blue). Eight mice per genotype were analyzed. Scale bars: 150 μm. Inset depicts a magnified view.

## Discussion

Functional deletion of single LUBAC components leads to the development of inflammatory skin phenotypes of varying severity. While the skin of *Sharpin$^{cpdm/cpdm}$* mice exhibits overall inflammation resembling atopic dermatitis, mice that lack HOIP or HOIL-1 selectively in keratinocytes develop a more severe cutaneous inflammation that results in early postnatal lethality[8,10,30,31]. Here, we show that mice lacking OTULIN selectively in keratinocytes exhibit a severe skin inflammation that presents on their tail skin and delineated regions of the back skin. These inflammatory lesions develop into verrucous carcinomas, an uncommon variant of squamous cell carcinomas, that are characterized by exophytic epidermal outgrowths and marked melanophagy, a phenotype that was not observed in mice lacking components of the LUBAC complex. Also, the degree of cutaneous inflammation observed in $\Delta^{\text{Ker}}$OTULIN mice was markedly milder than the severe skin phenotype observed in mice lacking HOIP or HOIL-1 in keratinocytes, which leads to early postnatal lethality[8]. Although $\Delta^{\text{Ker}}$OTULIN skin and keratinocytes showed a significant reduction in expression of LUBAC proteins, still residual LUBAC activity in the skin of these mice can be expected.

$\Delta^{\text{Ker}}$OTULIN mice are fully protected from dermatitis and skin tumorigenesis when crossed to a TNFR1-deficient or RIPK1 kinase-mutant background. This is in agreement with the phenotype in *Sharpin$^{cpdm/cpdm}$* mice that also do not develop skin inflammation in the absence of TNFR1 or RIPK1 kinase signaling[30,31], but is in contrast to the phenotype of keratinocyte-specific HOIP or HOIL-1 knockout mice that only show a delayed dermatitis in TNFR1 deficient or RIPK1 kinase-dead conditions[8]. Also, in contrast to primary keratinocytes from *Sharpin$^{cpdm/cpdm}$* mice that are highly sensitive to TNF-induced cell death, OTULIN-deficient keratinocytes are equally resistant to TNF-induced cell death as control keratinocytes. However, when primary keratinocyte cultures were primed with IFN-γ, $\Delta^{\text{Ker}}$OTULIN keratinocytes were sensitized to TNF-induced cell death. No differences, however, could be observed in TNF-induced NF-κB and MAPK responses between control and $\Delta^{\text{Ker}}$OTULIN primary keratinocytes.

The observation that genetic deletion of both FADD and MLKL in keratinocytes protects $\Delta^{\text{Ker}}$OTULIN mice from dermatitis development, proved that cell death of keratinocytes is the driving force of the cutaneous inflammation developing in $\Delta^{\text{Ker}}$OTULIN mice. MLKL deficiency could ameliorate but not prevent dermatitis development in $\Delta^{\text{Ker}}$OTULIN mice, suggesting that both FADD-dependent apoptosis and MLKL-dependent necroptosis are driving the skin lesion development in $\Delta^{\text{Ker-}}$OTULIN mice. Moreover, our data point to keratinocyte cell death preceding inflammation, as we observed apoptotic keratinocytes in $\Delta^{\text{Ker}}$OTULIN skin sections at a time-point (P0.5) when skin lesions were not apparent yet. Also, the absolute

number of infiltrating immune cells was still largely comparable in non-lesional $\Delta^{\text{Ker}}$OTULIN and control skin in these newborn mice, and no increase in epidermal thickness could be observed, again suggesting that keratinocyte death occurs prior to the inflammation. Our scRNAseq data also pointed out that several subsets of HFSCs expand in a progressive manner in non-lesional and lesional $\Delta^{\text{Ker}}$OTULIN skin, while other subsets gradually decrease in numbers. Whether these changes in stem cell lineage are due to changes in stem cell death rates or changes in proliferation under influence of OTULIN-deficiency, remains to be elucidated. This withstanding, it is clear that changes in keratinocyte stem cell fates occur prior to the substantial immune infiltration that is present in $\Delta^{\text{Ker}}$OTULIN skin lesions.

We were able to identify an important role for type 1 IFN signaling in mediating skin inflammation in $\Delta^{\text{Ker}}$OTULIN mice. Indeed, IFNAR1 deficiency rescued $\Delta^{\text{Ker}}$OTULIN mice from dermatitis development in about 60% of the mice. These findings agree with previous studies pinpointing a role for type-1 IFNs in OTULIN deficient or mutant mice[4,17,24], but implicate that other cytokines are also involved in regulating cutaneous inflammation in $\Delta^{\text{Ker}}$OTULIN skin. However, it should be noted that the IFNAR1 knockout mice have various immune defects[41], and were shown to be resistant to imiquimod-induced skin inflammation[42,43]. Prediction of ligand-target cell interactions, by combining single-cell expression data with prior knowledge on signaling and gene regulatory networks, allowed us to also identify IL1β as an important cytokine involved in the pathology of $\Delta^{\text{Ker}}$OTULIN mice. Consequently, pharmacological inhibition of IL1β signaling suppressed dermatitis development in $\Delta^{\text{Ker}}$OTULIN mice. Inflammatory symptoms in ORAS patients can be managed by neutralization of TNF[15,16,29], but IL1β neutralization by Anakinra showed efficacy in a patient with panniculitis and dermatosis[16], in agreement with our findings in mice.

OTULIN deficiency has distinct biological effects in different cellular and tissue contexts. OTULIN deletion in macrophages was shown to induce systemic inflammation in mice, while deletion of OTULIN in B- or T-cells did not result in overt phenotypes[15]. We and others recently showed that selective ablation of OTULIN in hepatocytes (OTULIN$^{\text{LPC-KO}}$) results in severe liver disease characterized by fibrotic and neoplastic responses[24,28]. While FADD deficiency or RIPK1 kinase-dead expression prevented hepatocyte cell death and development of liver disease in OTULIN$^{\text{LPC-KO}}$ mice, genetic deletion of TNF or TNFR1 did not ameliorate the liver phenotype[24], in sharp contrast with our observations in keratinocyte-specific OTULIN-deficient mice, where ablation of TNFR1 signaling resulted in a complete rescue of cutaneous inflammation. These data indicate that OTULIN regulates inflammatory cell death pathways with different modes of action, depending on the cellular context. One possible explanation for this is the difference in *Otulin* expression levels across different tissues. Indeed, analysis of gene expression

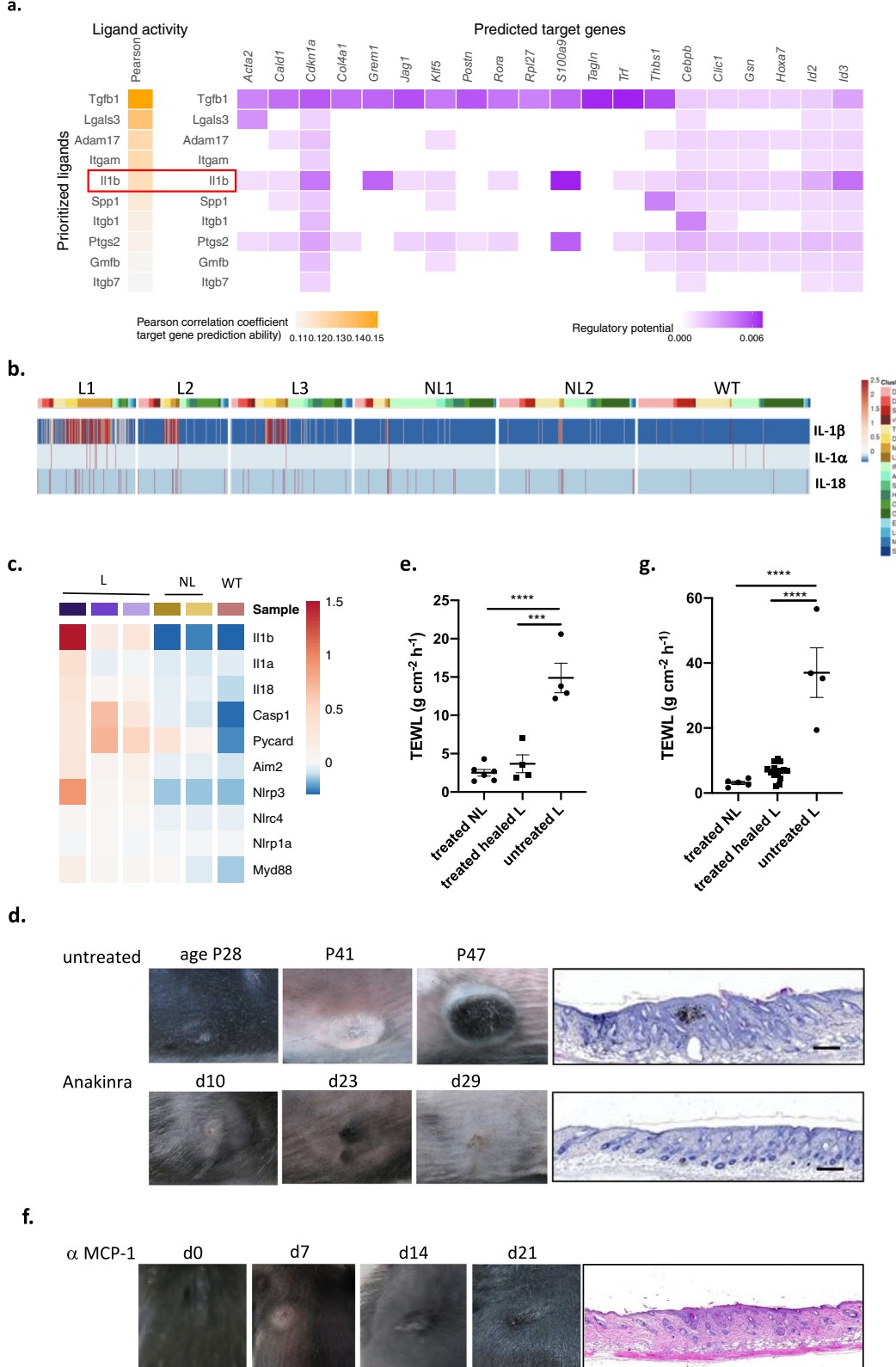

in different mouse tissues using the EBI expression atlas shows that *Otulin* expression strongly differs between different tissues with high expression in skin (Supplementary Fig. 7).

In conclusion, we have shown that linear deubiquitination of proteins by OTULIN serves as a crucial biological mechanism important for the maintenance of skin stem cell homeostasis and the prevention of keratinocyte death and subsequent skin

inflammation. This further demonstrates that aberrant cell death can act as the driving force for tissue inflammation and neoplastic responses. In line with our observations, a study by the group of Manolis Pasparakis[44], published in this issue of *Nature Communications*, describes similar findings, confirming the role of OTULIN in preventing skin inflammation by inhibiting the death of keratinocytes.

**Fig. 6 Interleukin-1β mediated signaling between innate immune cells and OTULIN-deficient keratinocytes regulates skin inflammation. a** Schematic representation of NicheNet analysis identifying ligands secreted by innate immune cells that bind to receptors on keratinocytes, mediating changes in gene expression. Lesional skin is compared to non-lesional Δ$^{Ker}$OTULIN skin. **b** Heatmap of IL-1β, IL-1α, and IL-18 mRNA expression levels in single cells of control (WT, $n = 1$), and Δ$^{Ker}$OTULIN non-lesional (NL, $n = 2$) and lesional (L, $n = 3$) skin, as determined by scRNAseq. **c** Heatmap of average mRNA expression levels of genes involved in inflammasome signaling in control, non-lesional Δ$^{Ker}$OTULIN and lesional Δ$^{Ker}$OTULIN keratinocytes, as determined by scRNAseq analysis. **d** Representative pictures of back skin of mice treated with daily injections of 300 mg/kg Anakinra and untreated controls. Age of the mice is indicated and duration of treatment. H&E-stained skin showing the morphology of the lesion at the time of sacrifice. Scale bar: 200 μm. **e** Trans-epidermal water loss (TEWL) measurements of non-lesional (NL) and healed lesional (L) skin of mice treated with Anakinra compared to untreated lesional skin (treated NL, $n = 5$; healed L and untreated L $n = 4$; ***$p < 0.001$; ****$p < 0.0001$, One-way ANOVA with multiple comparisons). Data represent means ± SEM. **f** Representative pictures of back skin of mice treated with intraperitoneal injections of 40 mg/kg α-MCP-1 antibody twice weekly. Duration of treatment is indicated. H&E-stained skin showing the morphology of the lesion at the time of sacrifice. Scale bar: 200 μm. **g** TEWL measurements of non-lesional (NL) and healed lesional (L) skin of mice treated with α-MCP-1 antibody compared to untreated lesional skin (treated NL, $n = 5$; healed L and untreated L $n = 4$; ***$p < 0.001$; ****$p < 0.0001$, One-way ANOVA with multiple comparisons). Data represent means ± SEM.

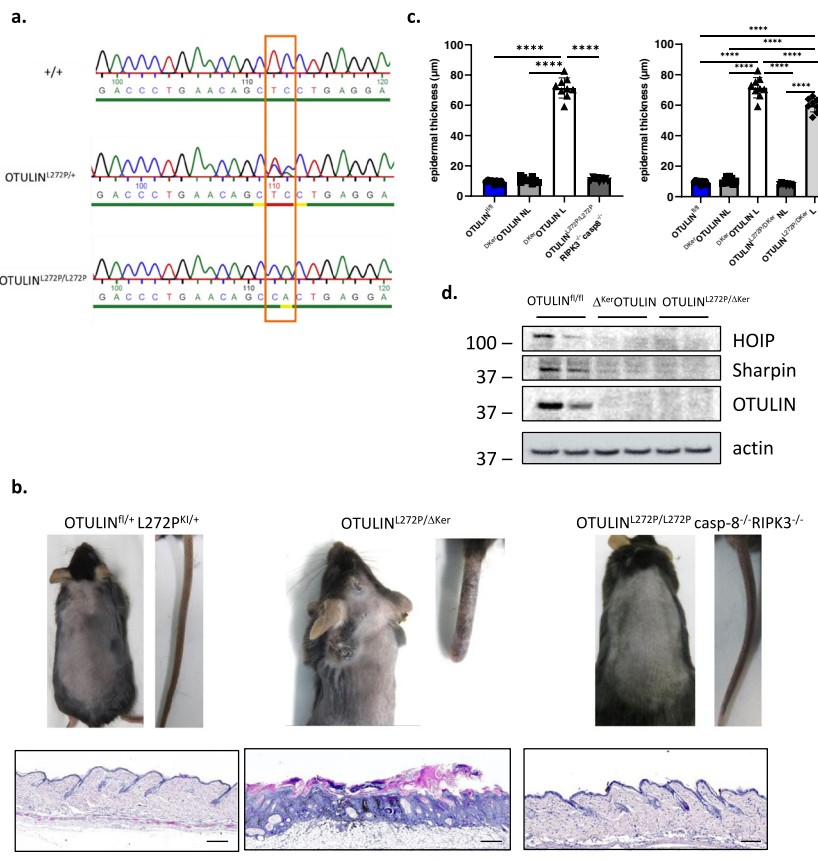

**Fig. 7 Knockin of a human mutation in the murine *Otulin* gene phenocopies OTULIN-deficiency. a** Targetting strategy for introduction of L272P mutation in the murine *Otulin* allele. **b** Representative images of the back skin and tail (upper panel), and H&E-stained skin sections (lower panel) of OTULIN$^{L272P/+}$, OTULIN$^{L272P/\Delta Ker}$, and OTULIN$^{L272P/L272P}$casp8$^{-/-}$RIPK3$^{-/-}$ mice. Scale bars: 200 μm. **c** Epidermal thickness quantification of skin of 7–11 weeks old mice. NL non-lesional; L lesional ($n = 10$ per condition; ****$p < 0.0001$, One-way ANOVA with multiple comparisons). Data represent means ± SEM. **d** Immunoblotting on epidermal tail lysates of 7-week old OTULIN$^{fl/fl}$ ($n = 2$), Δ$^{Ker}$OTULIN ($n = 2$) and OTULIN$^{L272P/\Delta Ker}$ ($n = 2$) mice using antibodies detecting HOIP, SHARPIN, and OTULIN. Anti-actin is shown as loading control. Molecular weight marker units are in kilodalton (kD).

## Methods

**Mice.** The following mouse lines were used: *Otulin*$^{FL24}$, *K14-Cre*[25], *Fadd*$^{FL45}$, *Mlkl*$^{FL46}$, *Ripk1*$^{D138N47}$, *Tnfr1*$^{-/-48}$, *Ifnar1*$^{-/-49}$, *IL1r*$^{-/-50}$, *Myd88*$^{-/-51}$, *casp8*$^{-/-52}$, and *Ripk3*$^{-/-53}$. All alleles were maintained on a C57BL/6 genetic background. Mice were housed in individually-ventilated cages at the VIB Centre for Inflammation Research, in a specific pathogen-free animal facility at 20−24 °C and 45−65% humidity with a 12 h light/dark cycle and food and water available at libitum. All experiments on mice were conducted according to institutional, national, and European animal regulations for animal testing and research. Animal protocols were approved by the VIB-Ghent University ethical review board.

**Generation of OTULIN$^{L272P}$ knockin mice.** For the generation of OTULIN$^{L272P}$ knockin mice, C57BL/6J zygotes were electroporated with 40 ng/μl Cas9 protein (iDT), 8 ng/μl cr/tracrRNA duplex (iDT) with guide sequence 5′ ATGACCCT

GAACAGCTCCTG 3′ and 150 ng/μl single-stranded DNA repair template with sequence 5′- AAGTGCCGTTCTTCTCTGTGCTCTTGTTTGCCCGAGACACA TCCAATGACCCTGAACAGCCACTGAGGAACCACCTAAACCAGGTGGGGAC ACACGGGGGGGCCTTGAGCAGGTGAGTTGTGGC-3′ (iDT) containing the L272P mutation (CTC > CCA). The embryos were transferred to foster mothers the same day through oviduct transfer. Effective mutagenesis was confirmed by sequencing of the *Otulin* gene on genomic DNA.

**Therapeutic treatment studies.** Δ$^{Ker}$OTULIN mice were either or not treated with 300 mg/kg Anakinra (Sobi, Kineret 100 mg/0.67 ml) by intraperitoneal (IP) injection daily from the age of 18 days on and for a period of 30 days, or with 40 mg/kg anti-MCP-1 antibody (InVivoMab, BE0185) IP twice a week from the age of 18 days on and for a period of 30 days.

**Table 1. depicting the number of offspring from the indicated intercrosses (observed and expected ratios).**

| OTULIN$^{L272P/+}$ x OTULIN$^{L272P/+}$ | Expected | Observed (at birth) |
|---|---|---|
| OTULIN$^{+/+}$ | 25% (26−27) | 34% (31) |
| OTULIN$^{L272P/+}$ | 50% (53) | 66% (75) |
| OTULIN$^{L272P/L272P}$ | 25% (26−27) | 0% (0) |
| Total (observed) | 100% (106) | 100% (106) |

| OTULIN$^{L272P/+}$ Casp-8$^{-/-}$ RIPK3$^{-/-}$ x OTULIN$^{L272P/+}$ Casp-8$^{-/-}$ RIPK3$^{-/-}$ | Expected | Observed (at 8 weeks of age) |
|---|---|---|
| OTULIN$^{+/+}$ | 25% (13−14) | 22% (12) |
| OTULIN$^{L272P/+}$ | 50% (27) | 54% (29) |
| OTULIN$^{L272P/L272P}$ | 25% (13−14) | 24% (13) |
| Total (observed) | 100% (54) | 100% (54) |

**Wound healing assay**. Full-thickness wounds on the back skin of shaved mice were made by using an 8 mm punch biopsy needle (Stiefel Instruments) under analgesia and general anesthesia in 7 weeks-old transgenic and control littermates. Wound sizes were measured every other day by two independent researchers, who were blinded to group allocations.

**Primary keratinocytes**. Primary mouse keratinocytes were isolated from Δ$^{Ker}$-OTULIN and OTULIN$^{fl/fl}$ skin as previously described[54]. Briefly, shaved back skin was isolated, sterilized, and floated on 0.25% trypsin overnight. The epidermis was separated from dermis and cultured on confluent feeder cultures.

**Flow cytometry**. Immunophenotyping of mouse skin was performed on single-cell suspensions obtained following trypsin digestion for 1 h at 37 °C and subsequent digestion with collagenase type-1 (1.25 mg/ml; GIBCO), type-2 (0.5 mg/ml, Sigma) and type-4 (0.5 mg/ml, Sigma) for 30−45 min. Cells were stained with the following fluorochrome-linked antibodies, as listed in Supplementary Table 1. Prior to measuring, counting beads (Life Technologies) were added to the cells. Measurements were performed on a BD Fortessa 5-laser cytometer and analyzed using FlowJo 10.6.1. software (Tree Star).

**Immunofluorescence**. Dewaxed paraffin or frozen skin sections or horizontal tail wholemounts were labeled with respective antibodies listed in Supplementary Table 1. As secondary antibodies donkey-anti-mouse 488 AlexaFluor (1:2000) and goat-anti-rabbit DyLight 586 (1:2000) were used in combination with Dapi.

**Immunohistochemistry**. Dewaxed paraffin skin sections were subjected to heat-mediated antigen retrieval (citrate buffer; pH = 6) and labeled with Ki-67 Ab (1:1000; Cell Signaling Technology, D3B5, 12202) or cleaved caspase-3 Ab (1: 1000; Cell Signaling Technology, 9661). Slides were incubated with secondary antibody followed by avidin-biotin complexes and peroxidase activity was detected with diaminobutyric acid (DAB) substrate (Vector Laboratories).

**EdU tracing**. Mice were intraperitoneally injected with EdU (1.25 mg/kg in PBS; Click-IT EdU Alexa 488 Imaging Kit) 3 h prior to sacrifice. Staining was performed according to the manufacturers' instructions.

**Transepidermal water loss (TEWL) measurements**. Mice were shaved one day prior to TEWL assessment. TEWL was measured on the back skin of anesthetized mice using a TEWA meter (courage and Khazaka, TM 210).

**Western blot analysis**. Primary keratinocytes and skin tissue were homogenized using E1A lysis buffer (50 mM HEPES pH7.6; 250 mM NaCl; 5 mM EDTA; 0.5% NP40) and NP-40 (50 mM Tris-HCl, pH 7.6; 1 mM EDTA; 150 mM NaCl; 1% NP-40; 0.5% sodiumdeoxycholate; 0.1% SDS) buffer respectively containing protease inhibitors (Roche) and phosphatase inhibitors (Sigma), denaturated in 1 × Laemmli buffer (50 mM Tris-HCl pH8.2; 2% SDS; 10% glycerol; 0.005% BFB; 5% β-mercapto-ethanol) and boiled for 10 min at 95 °C. 20 μg of liver lysates and 20 μg of cell lysates were separated by SDS-polyacrylamide gel electrophoresis (PAGE), transferred to nitrocellulose and analyzed by immunoblotting. Membranes were probed with antibodies against OTULIN (1:1000, Cell Signaling Technology, 14127), JNK (1:1000, BD Bioscience, 554285), phospho-JNK (1:1000, Millipore, PS1019), IκBα (1:1000, Santa Cruz Biotechnologies, sc371), phospho-IκBα (1:1000, Cell Signaling Technology, 9246), p38 MAPK (1:1000, Cell Signaling Technology, CST9212), phospho-p38 MAPK (1:1000, Cell Signaling Technology, CST9215), HOIL-1 (1:2000, kind gift of Dr. Henning Walczak, UCL London), HOIP (1:1000, kind gift of Dr. Rune Damgaard, MRC Cambridge), SHARPIN (1:1000, Proteintech, 14626-1-AP), Ripk1 (1:2000, Cell Signaling Technology, 3493), linear ubiquitin (1:2500, Millipore, clone LUB9, MABS451), caspase-3 (1:1000; Cell Signaling, 9662) and actin-HRP (1:10000, MP Biomedicals). As secondary antibodies, HRP coupled anti-rabbit-HRP, anti-mouse-HRP, and anti-goat-HRP were used (1:2500, Amersham) and detection was done by chemiluminescence (Western Lightning Plus ECL, Perkin Elmer) using the Amersham Imager 600 (GE Healthcare).

**Immunoprecipitation**. Recombinant GST-UBAN was produced in BL21(DE3) cells. In brief, BL21(DE3) cells were transformed with the plasmid encoding GST-UBAN and protein expression was induced with 0.5 M IPTG. After 4 h, cells were collected and lysed in lysis buffer (20 mM Tris-HCl pH 7.5, 10 mM EDTA, 5 mM EGTA, 150 mM NaCl, 1 mM DTT supplemented with phosphatase and protease inhibitor cocktail tablets (Roche Diagnostics)), sonicated, and cleared by centrifugation. After centrifugation, Triton-X100 (0.5% final concentration) was added to the supernatant, which was then transferred onto prewashed glutathione beads and left rotating for 2 h at 4 °C. After incubation, the beads were centrifuged, washed twice with washing buffer (20 mM Tris-HCl pH 7.5, 10 mM EDTA, 150 mM NaCl, 0.5% Triton-X100) and resuspended in resuspension buffer (20 mM Tris-HCl pH 7.5, 0.1% β-mercaptoethanol, 0.05% sodiumazide), ready to be used. Cell lysates from total skin tissue were prepared as described before, protein concentration was determined and 800 μg protein lysate was incubated overnight with GST-UBAN-containing glutathione beads. The next day, the beads were washed three times in RIPA lysis buffer (150 mM NaCl, 1% NP-40, 0.5% Sodium Deoxycholate, 0.1% SDS, 10 mM Tris-HCl pH 8 supplemented with phosphatase and protease inhibitor cocktail tablets (Roche Diagnostics)). Beads were then resuspended in 60 μL 1× laemmli buffer for direct analysis.

**Real-time RT PCR**. Total RNA was isolated using TRIzol reagent (Invitrogen) and Aurum Total RNA Isolation Mini Kit (Biorad), according to the manufacturer's instructions. Synthesis of cDNA was performed using Sensifast cDNA Synthesis Kit (Bioline) according to the manufacturer's instructions. cDNA was amplified on quantitative PCR in a total volume of 5 μl with SensiFAST SYBR® No-ROX Kit (Bioline) and specific primers on a LightCycler 480 (Roche). The reactions were performed in duplicates. The mouse-specific primers used are summarized in Supplementary Table 2.

**Cytokine detection**. Cytokine concentrations in serum were determined by a magnetic bead-based multiplex assay using Luminex technology (BioRad), according to the manufacturer's instructions. Cytokine concentrations observed in OTULIN$^{fl/fl}$ and Δ$^{Ker}$OTULIN mice were re-used in figure panels 1g, 3b, and 5e.

**scRNA sequencing and analysis**. Single-cell suspensions were obtained from total mouse skin as previously described[55]. Live cells were sorted on FACS Aria by gating for L/D eFluor780 negative cells into PBS with 0.04% BSA, spun down, and resuspended in PBS with 0.04% BSA at a final concentration of 1000 cells/μl. Cellular suspensions (target recovery of 10,000 cells) were loaded on a GemCode Single-Cell Instrument (10x Genomics, Pleasanton) to generate single-cell Gel Bead-in-Emulsion (GEMs). Single-cell RNA-Seq libraries were prepared using GemCode Single-Cell V2 3'Gel Bead and Library kit (10x Genomics) according to the manufacturer's instructions. Sequencing libraries were loaded at 2.1 pM loading concentration on a HiSeq4000 with custom sequencing metrics (single-indexed sequencing run, 28/8/0/98 cycles for R1/i7/i5/R2) (Illumina, San Diego, CA). Sequencing was performed at the VIB Nucleomics Core (VIB, Leuven, Belgium). Demultiplexing of the raw data and mapping to the mouse genome mm 10 (1.2.0) was done by the 10X CellRanger software (version 2.1.1; cellranger). Preprocessing of the data was done by the scran and scater R package (version 4.1) according to the workflow proposed by the Marioni and Theis lab[34]. Outlier cells were identified based on 3 metrics (library size, number of expressed genes, and mitochondrial proportion) and cells were tagged as outliers when they were a certain median absolute deviation (MAD) away from the median value of each metric across all cells. For the library size a lower bound of 4 and upper bound of 3 MADs were used, with exception of samples NL1, L3 (upper bound of 2,5 MADs) and WT (upper bound of 4 MADs). For the number of expressed genes the value was set to 4 MADs for both the upper and lower bound, with exception of samples L1, L2 (upper bound of 3 MADs) and L3 (upper bound of 2 MADs). For the mitochondrial proportion 10 MADs were used in samples L1, L2, and NL1, 15 MADs in L3 and 8 MADs in WT. Low-quality cells (low UMI counts, high percentage of mitochondrial genes) were removed from the analysis. A principal component analysis (PCA) plot was generated as an extra filtering using the runPCA function from the scater R package with the default parameters. Genes expressed in less than 3 cells and cells expressing less than 200 genes were removed. The samples were

aggregated using the merge function, counts were normalized and log2 transformed using the NormalizeData function, both from the Seurat R package (v3.1.0) using default parameters. Detecting highly variable genes, finding clusters, and creating UMAP plots was done using the Seurat pipeline. Clustering was performed using the first 34 principal components and a resolution of 0.8.

**Statistics**. Results are expressed as means ± SEM. Statistical significance between OTULIN$^{fl/fl}$ and $\Delta^{Ker}$OTULIN was assessed using a Mann−Whitney testing, One-way ANOVA, or Two-way ANOVA with multiple comparisons. Statistical significance between $\Delta^{Ker}$OTULIN and the different genetic crosses was assessed using a one-way ANOVA test, followed by Tukey's multiple comparison test. The analysis was performed with Prism 9 software. To compare the percentages of sytox green positive cells, we analyzed repeated measurements using the method of residual maximum likelihood (REML), as implemented in Genstat version 21. When representative images are shown, a minimum of 12 mice of the relevant genotype were analyzed. Comparison of HFSC marker positive cells in the permanent epidermis was done by Chi-square testing, comparing L or NL gene frequencies to expected frequencies from WT.

**Reporting summary**. Further information on research design is available in the Nature Research Reporting Summary linked to this article.

## Data availability

Source data are provided with this paper. The accession number for the raw scRNA-sequencing data reported in this study is Gene Expression Omnibus (GeO): GSE162394. Source data are provided with this paper.

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

## Acknowledgements

We thank the EUCOMM Consortium for *Otulin*-targeted ES cells, Manolis Pasparakis for providing RIPK1$^{D138N}$ mice, Vishva Dixit and Genentech for providing RIPK3$^{-/-}$ mice, and Alexander Warren and James Murphy (The Walter and Eliza Hall Institute of Medical Research, Melbourne, Australia) for the use of floxed *Mlkl* mice. We thank Sofie De Schepper for pathological assessment of Δ$^{Ker}$OTULIN mice, and Laetitia Bellen, Dimitri Huygebaert and Dieter Vanhede for animal care. We acknowledge helpful input from Jonathan Maelfait and the VIB BioImaging Core, the VIB Nucleomics Core, and the VIB Flowcore for technical assistance, and all members of the van Loo lab for suggestions and discussions. E.H. is supported by an FWO postdoctoral fellowship and a CRIG Young-Investigator grant. A.M. is supported by an FWO postdoctoral fellowship, L.v.H and L.V. are supported by an FWO FR fellowship. K.R. holds an Odysseus Grant and ERC Advanced grant. Research in the van Loo lab is financed by research grants from the FWO, Stichting Tegen Kanker, the Charcot Foundation, Kom op Tegen Kanker and the "Concerted Research Actions" (GOA) of the Ghent University.

## Author contributions

E.H.: conception and design of experiments, analysis and acquisition of data, drafting and revising the paper. K.L.: experimental design, analysis and acquisition of data, drafting and revising the paper. K.A., N.v.D., J.R., M.S., A.M., L.v.H., L.V., S.M., K.V., H.-K.V., K.R., Y.S., and M.K.: analysis and acquisition of data. T.H.: generation of OTULIN$^{L272P}$ knockin mice. G.v.L.: conception and design of experiments, data analysis, drafting, and revising the paper.

## Competing interests

The authors declare no competing interests.
