## [Peer Review File · Nature Communications]

REVIEWER COMMENTS

Reviewer #1 (Ubiquitin, LUBAC) (Remarks to the Author):

This manuscript by Hoste et al. investigates the role of the deubiquitinase OTULIN, which exclusively hydrolyzes linear ubiquitin chains, in the homeostatic maintenance of the skin. Mice with keratinocyte specific deletion of OTULIN were generated by crossing the K14-Cre with OTULIN^{f/f} mice. The authors show that the progeny of these mice, termed Δ KerOTULIN, develop severe skin inflammation as early as postnatal day 6 that progresses to verrucous carcinoma and is accompanied by systemic inflammation. The Δ KerOTULIN mice also have compromised skin barrier integrity and increased skin cell death. The full ablation of TNFR1, inhibition of kinase activity of RIPK1 or keratinocyte ablation of MLKL and FADD abrogates the skin inflammatory phenotype of the Δ KerOTULIN mice. Using single cell RNA sequencing, the authors examine the different populations of cells from the skin of wildtype mice and lesional or non-lesional skin of Δ KerOTULIN and identify IFN and IL-1b signaling signatures in the inflamed skin samples. They demonstrate that IFNAR1^{-/-} x Δ KerOTULIN mice were less susceptible to skin inflammation and that treatment of Δ KerOTULIN mice with anti-IL-1b antibody ameliorates the skin inflammation. Finally, the authors generate a mouse carrying a OTULIN point mutation (OTULINL272P) equivalent to a loss-of-function mutation previously found in human patients with ORAS. When bred to homozygosity, the mutation result in embryonic lethality reminiscent of OTULIN full knockout and when crossed to Δ KerOTULIN mice, the progeny (OTULINL272P/ Δ Ker) exhibits an inflammatory skin phenotype similar to the Δ KerOTULIN mice. The authors conclude that OTULIN has a paramount role in restraining skin inflammation and immune homeostasis.

Overall, the manuscript is well-written and the study describes a new role for OTULIN in the homeostasis of skin cells. The experiments appear well-reasoned and well-designed and provide mechanistic insight into most observed phenotypes through a range of crosses to knockout mice. For the most part the conclusions of the manuscript are well supported by the results. However, there are several points that need to be addressed to solidify the findings of the paper.

Major points

1. Increased cell death and reduced NF- κ B activation in response to TNF in keratinocytes and other cell types is a feature of LUBAC-deficiency (e.g. Gerlach et al. 2011, Ikeda et al. 2011, Rickard et al. 2014, Taraborrelli et al 2018). However, biochemical analysis of TNF pathway signalling in OTULIN-deficient keratinocytes showed no changes to NF- κ B or MAPK activation and there was no increased sensitization of primary keratinocyte death. Would this not suggests that the observed phenotypes are not a direct consequence of altered TNFR1 signalling, although the TNF-TNFR1 pathway clearly drives pathological inflammation? The authors should discuss this, particularly with regards to other aberrantly activated inflammatory pathways in the OTULIN-deficient keratinocytes, such as the type-I IFN or IL-1b, which could be exacerbated by TNF signalling. Related to this, deletion of TNFR1 in Δ KerOTULIN mice resulted in substantial reduction in production of inflammatory cytokines, including TNF. Would this not suggest that TNF-induced cytokine production and inflammation contributes to the skin pathology, possibly by stimulating in influx of activated macrophages and neutrophils, which in turn cause tissue damage and TNF-driven cell death and dermatitis.
2. Figure 1b: The images from the different genotypes should be the same magnification. Further, the figures (Fig1 A, B, C, D, H; Fig2 A, B; Fig4 C) are disproportionately stretched and seemingly elongated, which results in distortion particularly for histology and IF images.
3. In figures 1h, 2b, 5e and supplementary 3b, it appears that the same set of serum cytokine

measurements from the *Otulin*^{f/f} and Δ KerOTULIN mice are used for comparison with other genotypes. Re-use of data in multiple figures needs to be clearly stated. Secondly, this could give misleading results as the samples that are compared are from mice that are not littermates and possibly are from different genetic backgrounds. Also, is the sex and age of mice or time of collection of samples comparable? All these factors could influence the serum cytokine measurements.

5. The level of LUBAC components is reduced substantially in PMKs from Δ KerOTULIN mice as has been observed in other studies. The authors state this is due to proteasomal degradation but do not provide evidence for ubiquitination of LUBAC components or stabilisation by inhibiting the proteasome. It is important to include these experiments to demonstrate that it is indeed due to proteasomal degradation and not other mechanisms.

6. In figure 3d, there is a strong OTULIN band in the Δ KerOTULIN mice lanes, though it's of lower intensity than in the control mice bands. Is it plausible that the residual level of OTULIN in the cells is sufficient to maintain the near normal level of NF- κ B and MAPK signaling? Is this level of OTULIN in PMK cells from Δ KerOTULIN mice reproducible? Also, the authors do not state the n number for this panel in the figure legend.

7. It is unclear in the methods what skin layer have been used to prepare single cell suspensions for the scRNA-seq and the FACS analysis, this should be clarified. Further, supplementary figure 4b shows a surprising lack of T cells in their skin preps, except for gdT cells while the scRNA-seq data in figure 4a shows at least 3 clusters, within the T cell population, which are distinct between WT, NL and L samples. This is seemingly contradictory, and the authors should address this. Are all of these gdT cells or different subsets of T cells? It would also be interesting to shed some light on the transcriptomic alteration of the T cells in the WT, NL or L samples.

8. The authors convincingly show that Δ KerOTULIN x IFNAR1^{-/-} mice are protected from the skin inflammation induced by the lack of OTULIN in keratinocytes. However, it should be noted that the IFNAR mice are known for their various immune defects, including resistance to skin inflammation. Gui et al. 2016 (<https://doi.org/10.1016/j.jid.2016.06.608>) have shown severely reduced recruitment of hematopoietic cells to the skin of IFNAR mice following imiquimod (IMQ) challenge and lack of inflammatory cytokines in the skin following IMQ challenge. Therefore, the findings from the Δ KerOTULIN x IFNAR1^{-/-} mice should be discussed in the context of IFNAR1 knockout mice and their inherent defects, such as dysregulated immune system development (Gough et al., 2012; <https://doi.org/10.1016/j.immuni.2012.01.011>)

If possible, neutralization of IFNAR-1 using antibody treatments in the Δ KerOTULIN mice could be considered as an alternative to determine the role of type 1 IFN signalling in alleviating the Δ KerOTULIN skin inflammation. This experiment would provide more direct evidence.

9. In relation to figure 5f, it is stated in lines 283-285 that the IFN- β staining in normal skin sections is probably due to dendritic cells and cite reference #35. However, this is not backed up by experimental evidence. The authors cite the early work by Wollenberg et al., 2002, which looked at plasmacytoid dendritic cells not classical dendritic cells, in the human skin. The authors should therefore either provide relevant reference, experimental evidence that backs up this claim, or omit this statement. Nonetheless, it's valuable if the authors could at least provide co-staining for CD45 to determine the source of IFN β .

10. As the OTULIN^{L272P}/ Δ Ker is a new mouse model, the authors should provide some further characterization. Specifically, biochemical analysis of keratinocytes from these mice such as protein abundance of OTULIN, LUBAC and linear Ub, and TNF-induced NF- κ B and MAPK signalling should be included .

Specific/minor points

1. In figure 1f and in the text lines 113-114, the authors use F4/80 staining to indicate the infiltration of macrophages in the non-lesional and lesional skin of Δ KerOTULIN mice. Admittedly, the different

populations of myeloid cells in the mouse skin have overlapping cell surface markers, making their identification difficult. However, F4/80 alone is insufficient to identify macrophages, as it has been reported that eosinophils, which the authors show are increased in NL skin in figure 4d, can express various levels of F4/80 in different tissues. (<https://doi.org/10.1038/cmi.2010.31>; <https://doi.org/10.1186/bcr441> ; DOI: 10.1038/s41467-018-06316-9 and <https://doi.org/10.1016/j.jaci.2012.07.025>). It would be worthwhile co-staining with an eosinophile marker such as Siglec-F, as the authors did for their flow cytometry analysis, or CD64 and CD11b to confirm that these indeed are macrophages.

2. In figure 1h and supplementary figure 1d the authors show that Δ KerOTULIN mice exhibit systemic inflammation with elevated serum IL-6, TNF, MCP-1 and IL-17 as well as enlarged inguinal lymph nodes. An increased infiltration of F4/80+ cells in the skin of Δ KerOTULIN mice along with upregulation of Il6 and Tnf in the skin of Δ KerOTULIN mice is shown (figure 1f), but the expression of Ccl2 (MCP-1) in the skin of Δ KerOTULIN mice is not addressed. CCL2 is a known chemoattractant for various immune cells including monocytes and it would therefore be relevant to determine the levels of Ccl2 gene expression in the skin as it CCL2 might be responsible for recruitment of monocytes and the increase in F4/80+ cells? Related to this, it would be interesting to investigate whether the blockade of monocyte migration, for instance with a-CCL2 treatment, could alleviate the skin lesions or the systemic inflammation, particularly in light of the results shown in figure 5 that neutralization of IL-1b, which is produced mostly by skin-infiltrating macrophages, alleviates skin inflammation in the Δ KerOTULIN mice.

3. Figure 1C. Include labels for images.

4. Also, the M1 Ub smears shown in the same figure are not smears but rather distinct demarcated bands. Indeed, in the Δ KerOTULIN samples it seems that there's a distinct increase in the M1 Ub bands at approximately 60 and 70 kDa. Do the authors know why this is the case?

5. In figure 1i, there is a distinctive pattern to the Ikba and p-Ikba bands in the second lane of the Δ KerOTULIN mice. This could happen if the membrane was stripped and re-probed. However, as Ikba and p-Ikba would run very closely on the gel (1 kDa difference), it would be hard to distinguish Ikba and p-Ikba particularly if there was residual primary antibody on the membrane following the stripping.

6. The authors nicely show that the ablation of FADD and MLKL in keratinocytes that lack OTULIN inhibits the skin inflammation in figures 3a. However, the combined loss of FADD and MLKL does not address whether its apoptotic or necroptotic cell death. It would be beneficial to show the data from the individual FADD and MLKL crosses with Δ KerOTULIN mice. Further, immunoblotting for markers of active apoptosis and necroptosis (e.g., p-MLKL, p-RIPK3, p-RIPK1) in the skin of Δ KerOTULIN mice and the changes thereof in the FADD/MLKL crosses, is relevant for the mechanistic understanding of the inflammatory phenotype of the Δ KerOTULIN mice.

7. Do the Δ KerOTULIN mice lose weight as a result of their skin lesions?

8. Figure 1b is missing annotation atop of the images, i.e., lesional vs non-lesional. Also, for figure 1c, please show H&E stains, at the same resolution, from non-cancerous Δ KerOTULIN skin and from control mice.

9. Figure 1f: please provide isotype staining controls, particularly for F4/80 staining. In the middle panels, is the Filagrin staining from control mice or non-lesional Δ KerOTULIN mice? please show both. Also, if possible, staining for total CD45+ cells is a valuable control to show and would nicely complement the data in this figure as well as in figure 4.

10. Supplementary figure 1c: inconsistent resolution of the images (indicated by different sized scale bars)

11. Figure 1g: how does this compare to tail skin lysates? lesions vs non-lesional skin?

12. Figure. 2a: scale bar is missing from ki-67 Δ KerOTULIN NL panel.

13. Figure 2c: scale bars are missing from both lower panels.

14. Figure 2g: low resolution makes it harder to judge the staining of cleaved caspase-3.

15. Supplementary figure 2: in panel (a) the authors show the wound size measurements up until d12

pw, however in panel (c) they show H&E stains at d14 pw that shows complete wound healing in the Δ KerOTULIN mice, notwithstanding the formation of the tumour-like lesion. Thus, it would be useful if the authors could show the wound size measurements up until d14 pw in panel (a).

16. Figure 3a: please provide epidermal thickness measurements for the comparison between the different genotypes. Also, the H&E representative images are of different resolutions (as indicated by different scale bars), consider showing consistent images.

17. Line 269: "IFN-response genes (IRGs)" are more commonly known in the literature as Interferon-stimulated genes (ISGs). Consider changing.

18. Figure 5b: Why the choice to look at gene expression in epidermal tail lysates instead of back skin non-lesion and lesion from Δ KerOTULIN, which would be a better comparison to the scRNA-seq data?

19. Supplementary 2c and 3c: missing scale bars in some panels.

20. Figure 5c: The H&E panels are missing annotation and are of different resolution. Also, the third H&E panel depicts what looks like inflamed skin in the Δ KerOTULIN x IFNAR1^{-/-} mice therefore please provide epidermal thickness measurements across genotypes for comparison.

21. Figure 7c: right panel in the H&E staining is missing scale bar.

22. Please provide clone and catalogue number details for all antibodies used in the manuscript.

Reviewer #2 (Innate signaling, ubiquitination, NFkB) (Remarks to the Author):

In this manuscript, authors report that OTULIN is a crucial regulator for maintaining skin cell homeostasis and preventing keratinocyte death and subsequent skin inflammation. OTULIN deletion in keratinocytes results in enhanced keratinocyte proliferation and cell death. TNFR1 deficiency or knockin expression of kinase-inactive RIPK1 or FADD and MLKL deletion prevents dermatitis development in the keratinocyte-specific OTULIN-deficient mice. In addition, the authors show that type 1 IFNs and IL-1b contribute to the skin inflammation in keratinocyte-specific OTULIN-deficient mice. These findings are interesting and provide new insight into this field; however, the current version of the manuscript has several issues that need to be addressed to strengthen the conclusions. Major comments:

1. Fig.1i shows increased NF-kB activity in the epidermal tail lysates of OTULIN deficient mice.

However, OTULIN deficiency has no effect on TNF-induced NF-kB signaling in primary keratinocytes (Fig. 3d). The authors should discuss the potential mechanism by which OTULIN regulates keratinocyte signaling in vivo. Does OTULIN deficiency in primary keratinocytes promote induction of NF-kB and MAPK signaling by proinflammatory cytokines, such as IL-1b?

2. Lesional skin of Δ KerOTULIN mice have enhanced cell proliferation and cell death (Fig. 2).

However, it is unclear whether the cells positive for cleaved caspase 3 and Ki67 are keratinocytes or infiltrating immune cells. Also, the authors should discuss how OTULIN deficiency increases keratinocyte proliferation.

3. Regarding Fig.2, what is the phenotype of keratinocyte proliferation and epidermal thickness in Δ Ker OTULIN-TNFR1^{-/-}, Δ Ker OTULIN-RIPK1D138N/D138N, Δ Ker OTULIN/FADD/MLKL mice? Also, mechanistically, what is the relationship between aberrant cell death and cell proliferation in Δ Ker OTULIN mice?

4. It is interesting that OTULIN deficiency has no effect on TNF-induced cell death in primary keratinocytes (Fig. 3c). However, the data are not quite convincing, because TNF did not induce cell death in WT cells (untreated and TNF-treated cells had similar level of cell viability). Since TNFR1 deficiency rescues Δ Ker OTULIN mice from dermatitis development, it is important to repeat this experiment using a higher dose of TNFa. If OTULIN deficiency indeed has no effect on TNF-induced cell death in primary keratinocytes, the authors should discuss how OTULIN may regulate cell death in

vivo. Which cell death trigger could be regulated OTULIN?

5. Fig. 3e shows a substantial reduction in the level of RIPK1 in Δ KerOTULIN keratinocytes. Is this result reproducible? Does OTULIN regulate RIPK1 stability?

6. The finding that OTULIN regulates stem cell populations is interesting (Fig. 4f). Could the authors propose the possible signaling mechanism (which receptor pathway might be regulated by OTULIN)?

7. What is the potential mechanism by which OTULIN-deficiency in keratinocytes induce innate immune cells infiltration (Fig. 4)?

8. Fig.5: how does OTULIN regulate IFN and IFN-response genes? What's the phosphorylation level of TBK-1 and IRF3?

Point-by-point response to reviewer's comments

Reviewer #1:

“Overall, the manuscript is well-written and the study describes a new role for OTULIN in the homeostasis of skin cells. The experiments appear well-reasoned and well-designed and provide mechanistic insight into most observed phenotypes through a range of crosses to knockout mice. For the most part the conclusions of the manuscript are well supported by the results. However, there are several points that need to be addressed to solidify the findings of the paper.”

We thank this Reviewer for his/her positive assessment of our manuscript.

Major points

1. “Increased cell death and reduced NF- κ B activation in response to TNF in keratinocytes and other cell types is a feature of LUBAC-deficiency (e.g. Gerlach et al. 2011, Ikeda et al. 2011, Rickard et al. 2014, Taraborrelli et al 2018). However, biochemical analysis of TNF pathway signalling in OTULIN-deficient keratinocytes showed no changes to NF- κ B or MAPK activation and there was no increased sensitization of primary keratinocyte death. Would this not suggest that the observed phenotypes are not a direct consequence of altered TNFR1 signalling, although the TNF-TNFR1 pathway clearly drives pathological inflammation? The authors should discuss this, particularly with regards to other aberrantly activated inflammatory pathways in the OTULIN-deficient keratinocytes, such as the type-I IFN or IL-1b, which could be exacerbated by TNF signalling.”

Indeed, no increased sensitization of primary keratinocyte to TNF-induced cell death could be observed in OTULIN-deficient cultured PMKs. However, we now show that OTULIN deficient PMKs are sensitized to TNF-induced cell death when primed with IFN- γ prior to TNF-stimulation. We included these new findings in Figure 3 of the revised manuscript (Fig. 3d and e). No differences could be observed in TNF-induced NF- κ B and MAPK activation between OTULIN-deficient and -proficient PMKs. We now include qPCR data on supernatants demonstrating that TNF induces inflammatory cytokine and chemokine expression, but without significant differences in levels between the two genotypes (new Supplementary Figure 3d).

“Related to this, deletion of TNFR1 in Δ KerOTULIN mice resulted in substantial reduction in production of inflammatory cytokines, including TNF. Would this not suggest that TNF-induced cytokine production and inflammation contributes to the skin pathology, possibly by stimulating in influx of activated macrophages and neutrophils, which in turn cause tissue damage and TNF-driven cell death and dermatitis.”

This Reviewer correctly points out that stimulating the influx of activated macrophages and neutrophils contributes to TNF-driven dermatitis. We could now confirm that the chemokine MCP-1 is not only upregulated in skin of Δ^{Ker} OTULIN mice (as shown in a new Figure panel 1f), but also significantly contributes to the skin pathology. Indeed, we now show that intraperitoneal injection of Δ^{Ker} OTULIN mice twice a week with an MCP-1 blocking antibody could ameliorate the dermatitis in back skin, and completely rescue the inflammatory phenotype in Δ^{Ker} OTULIN tail skin (new Figure 6f-g and Supplementary Figure 6).

2. “Figure 1b: The images from the different genotypes should be the same magnification. Further, the figures (Fig1 A, B, C, D, H; Fig2 A, B; Fig4 C) are disproportionately stretched and seemingly elongated, which results in distortion particularly for histology and IF images.”

We included a new panel of images in Figure 1b for which all scale bars represent the same length. The lowest panel of Figure 1b depicts a magnified view of the verrucous carcinomas, with a differently sized scale bar as indicated in the figure legend. We have also included novel images in Figure 1a. We would like to point out that the total skin of Δ^{Ker} OTULIN mice is significantly thicker than normal skin and shows extensive induction of anagen, with highly elongated hair follicles. Therefore, it might seem that these microscopic images are stretched or elongated, while this is not the case. If images were adjusted for size, the scaling ratio (height versus width) was locked, thereby avoiding any distortion of the original images.

3. “In figures 1h, 2b, 5e and supplementary 3b, it appears that the same set of serum cytokine measurements from the *Otulin^{f/f}* and Δ^{Ker} OTULIN mice are used for comparison with other genotypes. Re-use of data in multiple figures needs to be clearly stated. Secondly, this could give misleading results as the samples that are compared are from mice that are not littermates and possibly are from different genetic backgrounds. Also, is the sex and age of mice or time of collection of samples comparable? All these factors could influence the serum cytokine measurements.”

The same set of serum samples from *OTULIN^{f/f}* and Δ^{Ker} OTULIN mice were used to show cytokine levels in different figure panels. This has now clearly been stated in the material and methods section. All mice were between 7 and 11 weeks old and kept on C57BL/6 genetic background and a similar ratio male and females was used. It is virtually impossible to always compare littermate samples when comparing different mouse lines with multiple mutant alleles.

5. “The level of LUBAC components is reduced substantially in PMKs from Δ^{Ker} OTULIN mice as has been observed in other studies. The authors state this is due to proteasomal degradation but do not provide evidence for ubiquitination of LUBAC components or stabilisation by inhibiting the proteasome. It is important to include these experiments to demonstrate that it is indeed due to proteasomal degradation and not other mechanisms.”

We now stimulated PMKs obtained from *OTULIN^{f/f}* and Δ^{Ker} OTULIN mice with TNF in the presence or absence of the proteasome inhibitor MG132. Immunoblot analysis on lysates from these cells indeed confirm that degradation of SHARPIN and HOIP is reduced in the presence of the proteasome inhibitor. These data are now shown in a new Supplementary Figure panel 3e.

6. “In figure 3d, there is a strong OTULIN band in the Δ^{Ker} OTULIN mice lanes, though it’s of lower intensity than in the control mice bands. Is it plausible that the residual level of OTULIN in the cells is sufficient to maintain the near normal level of NF-kB and MAPK

signaling? Is this level of OTULIN in PMK cells from Δ KerOTULIN mice reproducible? Also, the authors do not state the n number for this panel in the figure legend.”

There is indeed a residual OTULIN band visible in the western blot of Figure 3f (former panel 3d). This is due to the presence of feeder cells in the PMK cultures. PMKs used for these experiments were only passaged once prior to TNF stimulation, to avoid putative mutagenesis of the cells. If PMKs are passaged multiple times the contribution of the mitotically inactive feeders will be reduced, and OTULIN will no longer be observed in PMKs from Δ KerOTULIN mice (as shown in Supplementary Figure panel 1a). We now also state (in the figure legend) the number of independent experiments that have been performed.

7. “It is unclear in the methods what skin layer have been used to prepare single cell suspensions for the scRNA-seq and the FACS analysis, this should be clarified. Further, supplementary figure 4b shows a surprising lack of T cells in their skin preps, except for gdT cells while the scRNA-seq data in figure 4a shows at least 3 clusters, within the T cell population, which are distinct between WT, NL and L samples. This is seemingly contradictory, and the authors should address this. Are all of these gdT cells or different subsets of T cells? It would also be interesting to shed some light on the transcriptomic alteration of the T cells in the WT, NL or L samples.”

We used total skin for the scRNA-sequencing and flow cytometry analysis, which we now clearly state in the revised materials and methods section. The new Supplementary figure panels 4c and d show a subclustering of the T-cells identified by scRNA-seq analysis and Supplementary Figure 4e displays a frequency plot, showing the distribution over the different conditions. This subclustering revealed a substantial upregulation of Tregs in non-lesional and lesional Δ KerOTULIN skin, which we confirmed by flow cytometric analysis of different T cell populations on total skin of OTULIN^{fl/fl} and non-lesional and lesional Δ KerOTULIN skin. These results have been included in a new Supplementary Figure 4f and are discussed in the revised manuscript.

8. “The authors convincingly show that Δ KerOTULIN x IFNAR1^{-/-} mice are protected from the skin inflammation induced by the lack of OTULIN in keratinocytes. However, it should be noted that the IFNAR mice are known for their various immune defects, including resistance to skin inflammation. Gui et al. 2016 (<https://doi.org/10.1016/j.jid.2016.06.608>) have shown severely reduced recruitment of hematopoietic cells to the skin of IFNAR mice following imiquimod (IMQ) challenge and lack of inflammatory cytokines in the skin following IMQ challenge. Therefore, the findings from the Δ KerOTULIN x IFNAR1^{-/-} mice should be discussed in the context of IFNAR1 knockout mice and their inherent defects, such as dysregulated immune system development (Gough et al., 2012; <https://doi.org/10.1016/j.immuni.2012.01.011>). If possible, neutralization of IFNAR-1 using antibody treatments in the Δ KerOTULIN mice could be considered as an alternative to determine the role of type 1 IFN signalling in alleviating the Δ KerOTULIN skin inflammation. This experiment would provide more direct evidence.”

IFNAR1^{-/-} mice have indeed inherent immune defects which may obscure our observations, as remarked by the Reviewer. Hence, we now shortly refer to this issue in the discussion section of the revised manuscript. We also agree with the reviewer that a treatment protocol, using IFNAR1 neutralizing antibodies, would be interesting to directly address the importance of type I IFN signaling in the phenotype of Δ KerOTULIN mice. However, due to other priorities with the available mice we could not initiate this experiment.

9. “In relation to figure 5f, it is stated in lines 283-285 that the IFN- β staining in normal skin sections is probably due to dendritic cells and cite reference #35. However, this is not backed

up by experimental evidence. The authors cite the early work by Wollenberg et al., 2002, which looked at plasmacytoid dendritic cells not classical dendritic cells, in the human skin. The authors should therefore either provide relevant reference, experimental evidence that backs up this claim, or omit this statement. Nonetheless, it's valuable if the authors could at least provide co-staining for CD45 to determine the source of IFN β ."

We performed a double-staining of IFN- β and CD45, as suggested by the Reviewer, and included these images in a new Figure 5g. We also adapted the statement in the manuscript mentioning that '*Immunostaining for the type-1 interferon IFN- β revealed marked expression of this cytokine in lesional Δ^{Ker} OTULIN skin*', with no further details about the source of this interferon.

10. "As the OTULIN^{L272P}/ Δ Ker is a new mouse model, the authors should provide some further characterization. Specifically, biochemical analysis of keratinocytes from these mice such as protein abundance of OTULIN, LUBAC and linear Ub, and TNF-induced NF- κ B and MAPK signalling should be included."

Epidermal tail lysates of OTULIN^{L272P}/ Δ Ker mice were immunoblotted for expression of OTULIN, SHARPIN and HOIP. These experiments demonstrated that the L272P mutation renders the OTULIN protein unstable, confirming previous reports (Damgaard et al., 2016). We now include these new data in Figure 7e. Unfortunately, due to a shortage in OTULIN^{L272P}/ Δ Ker mice and technical issues with our primary keratinocyte cultures, we were not able to perform NF- κ B and MAPK signaling studies, as requested. However, since we confirmed that the L272P mutation results in a loss of OTULIN expression, we hypothesize that OTULIN^{L272P}/ Δ Ker PMKs will behave as Δ^{Ker} OTULIN PMKs, and will not show differences in TNF-induced NF- κ B and MAPK activation compared to control wild-type PMKs.

Specific/minor points

1. "In figure 1f and in the text lines 113-114, the authors use F4/80 staining to indicate the infiltration of macrophages in the non-lesional and lesional skin of Δ KerOTULIN mice. Admittedly, the different populations of myeloid cells in the mouse skin have overlapping cell surface markers, making their identification difficult. However, F4/80 alone is insufficient to identify macrophages, as it has been reported that eosinophils, which the authors show are increased in NL skin in figure 4d, can express various levels of F4/80 in different tissues. (<https://doi.org/10.1038/cmi.2010.31>; <https://doi.org/10.1186/bcr441> ; DOI: 10.1038/s41467-018-06316-9 and <https://doi.org/10.1016/j.jaci.2012.07.025>). It would be worthwhile co-staining with an eosinophile marker such as Siglec-F, as the authors did for their flow cytometry analysis, or CD64 and CD11b to confirm that these indeed are macrophages."

This Reviewer correctly points out that F4/80 can be expressed by other cell-types than macrophages in skin. Therefore, we performed a double staining with anti-F4/80 and anti-CD11b antibodies, and could demonstrate that F4/80+ cells are also expressing the CD11b marker in lesional KO skin, proving that there indeed is a marked infiltration of inflammatory macrophages. We have adjusted the text and inserted a novel panel in Figure 1e.

2. "In figure 1h and supplementary figure 1d the authors show that Δ KerOTULIN mice exhibit systemic inflammation with elevated serum IL-6, TNF, MCP-1 and IL-17 as well as enlarged inguinal lymph nodes. An increased infiltration of F4/80+ cells in the skin of Δ KerOTULIN mice along with upregulation of Il6 and Tnf in the skin of Δ KerOTULIN mice is shown (figure 1f), but the expression of Ccl2 (MCP-1) in the skin of Δ KerOTULIN mice is

not addressed. CCL2 is a known chemoattractant for various immune cells including monocytes and it would therefore be relevant to determine the levels of Ccl2 gene expression in the skin as CCL2 might be responsible for recruitment of monocytes and the increase in F4/80+ cells? Related to this, it would be interesting to investigate whether the blockade of monocyte migration, for instance with a-CCL2 treatment, could alleviate the skin lesions or the systemic inflammation, particularly in light of the results shown in figure 5 that neutralization of IL-1b, which is produced mostly by skin-infiltrating macrophages, alleviates skin inflammation in the Δ KerOTULIN mice.”

We now include qPCR data showing that MCP-1 (CCL2) is indeed upregulated in skin of Δ^{Ker} OTULIN mice (as shown in Figure panel 1f). In addition, as suggested by the Reviewer, we demonstrate that intraperitoneal injection of Δ^{Ker} OTULIN mice twice a week with an MCP-1 blocking antibody could ameliorate the dermatitis in back skin, and completely rescue the inflammatory phenotype in tail skin of Δ^{Ker} OTULIN mice (new Figure 6f-g and Supplementary Figure 6).

3. “Figure 1C. Include labels for images.”

We have incorporated the H&E sections of the verrucous carcinomas in a new Figure panel 1b, and labelled this panel accordingly.

4. “Also, the M1 Ub smears shown in the same figure are not smears but rather distinct demarcated bands. Indeed, in the Δ KerOTULIN samples it seems that there’s a distinct increase in the M1 Ub bands at approximately 60 and 70 kDa. Do the authors know why this is the case?”

Immunoblotting with antibodies against linear ubiquitin chains revealed the presence of increased amounts of M1-linked ubiquitin in epidermal lysates from Δ^{Ker} OTULIN mice (Figure 1h), in agreement with the function of OTULIN as an M1 ubiquitin-specific deubiquitinase. However, we do not know why these M1-ubiquitin chains occur as distinct bands rather than as a ubiquitin smear as we would expect from other studies. We can only speculate that the distinct bands visible on immunoblot are specific proteins labeled with M1 chains.

5. “In figure 1i, there is a distinctive pattern to the Ikba and p-Ikba bands in the second lane of the Δ KerOTULIN mice. This could happen if the membrane was stripped and re-probed. However, as Ikba and p-Ikba would run very closely on the gel (1 kDa difference), it would be hard to distinguish Ikba and p-Ikba particularly if there was residual primary antibody on the membrane following the stripping.”

We would like to point out that no membrane stripping was performed. Either different blots (with same lysates) were incubated with the different antibodies, or blots were first incubated with phospho-Ikba α antibodies and subsequently with an antibody against (non-phosphorylated) Ikba α . We have confirmed these data multiple times, allowing us to conclude that epidermal tail lysates from Δ^{Ker} OTULIN mice reveal an enhanced NF- κ B response compared to control epidermis.

6. “The authors nicely show that the ablation of FADD and MLKL in keratinocytes that lack OTULIN inhibits the skin inflammation in figures 3a. However, the combined loss of FADD and MLKL does not address whether its apoptotic or necroptotic cell death. It would be beneficial to show the data from the individual FADD and MLKL crosses with Δ KerOTULIN mice. Further, immunoblotting for markers of active apoptosis and necroptosis (e.g., p-MLKL, p-RIPK3, p-RIPK1) in the skin of Δ KerOTULIN mice and the changes

thereof in the FADD/MLKL crosses, is relevant for the mechanistic understanding of the inflammatory phenotype of the $\Delta^{Ker}OTULIN$ mice.”

We now include new findings showing that $\Delta^{Ker}OTULIN$ mice crossed onto an MLKL-deficient genetic background are partially protected from dermatitis. These compound transgenic $\Delta^{Ker}OTULIN/MLKL$ mice exhibit normal tails, but still develop sporadic lesions on their back skin (new Supplementary Figure 3c). As ablation of FADD in keratinocytes results in severe dermatitis and postnatal lethality in mice (Bonnet et al., 2011), we did not generate compound transgenic $\Delta^{Ker}OTULIN/FADD$ mice. Based on our observations in $\Delta^{Ker}OTULIN/FADD/MLKL$ (full protection) and $\Delta^{Ker}OTULIN/MLKL$ (partial protection), we conclude that both apoptosis and necroptosis of keratinocytes contributes to the dermatitis in $\Delta^{Ker}OTULIN$ mice. Unfortunately, due to priorities with the available mice, we were not able to perform the requested immunoblot experiments for markers of active apoptosis and necroptosis. Staining for cleaved caspase-3 on skin sections of $\Delta^{Ker}OTULIN/FADD/MLKL$ mice, however, revealed complete absence of apoptotic cells in the epidermis of these mice (Figure R1, for reviewers only).

Figure R1 : Immunohistochemical staining for cleaved caspase-3 in $OTULIN^{fl/fl}, \Delta^{Ker}OTULIN$ (non-lesional and lesional), and $\Delta^{Ker}OTULIN/MLKL/FADD$ mice. Cleaved caspase-3-positive cells are indicated by arrows.

7. “Do the $\Delta^{Ker}OTULIN$ mice lose weight as a result of their skin lesions?”

We monitored body mass of $\Delta^{Ker}OTULIN$ and control mice after weaning and could not observe weight loss in $\Delta^{Ker}OTULIN$ compared to control littermate mice, as shown in the figure below (Figure R2, for reviewers only). However, we only followed body weight for up to 9 weeks, since mice needed to be culled after for ethical reasons. Hence, it is plausible that $\Delta^{Ker}OTULIN$ mice would start losing weight at later age.

Figure R2 : body weight of Δ^{Ker} OTULIN and OTULIN^{fl/fl} littermate mice followed over time

8. “Figure 1b is missing annotation atop of the images, i.e., lesional vs non-lesional. Also, for figure 1c, please show H&E stains, at the same resolution, from non-cancerous Δ^{Ker} OTULIN skin and from control mice.”

We have adapted Figure 1b, as requested by the Reviewer.

9. “Figure 1f: please provide isotype staining controls, particularly for F4/80 staining. In the middle panels, is the Filaggrin staining from control mice or non-lesional Δ^{Ker} OTULIN mice? please show both. Also, if possible, staining for total CD45+ cells is a valuable control to show and would nicely complement the data in this figure as well as in figure 4.”

Staining with isotype control did not reveal any specific staining apart from the expected auto-fluorescence caused by hair follicles in the green channel (Figure R3, for reviewers only).

Figure 1e shows three panels for each of the stainings, one panel for OTULIN^{fl/fl} (left), one for Δ^{Ker} OTULIN non-lesional (NL, middle), and one for Δ^{Ker} OTULIN lesional (L, right) skin. We labeled these panels accordingly. We also included a double-staining for F4/80+ and CD11b+ cells, as requested by this Reviewer.

Figure R3 : isotype staining control of skin tissue sample.

10. “Supplementary figure 1c: inconsistent resolution of the images (indicated by different sized scale bars)”

We inserted same magnification views of the Oil red O stainings in Supplementary Figure 1c.

11. “Figure 1g: how does this compare to tail skin lysates? lesions vs non-lesional skin?”

(new Figure panel 1f) RT-qPCR analyses were performed on tail skin lysates, since in these type of lysates the epidermal fraction is far greater than in full skin lysates. Hence, epidermal transcriptional changes are better probed in tail skin lysates relative to back skin lysates. Since the total tail is affected in Δ^{Ker} OTULIN mice, no distinction can be made between lesional and non-lesional tail skin.

12. “Figure. 2a: scale bar is missing from ki-67 Δ KerOTULIN NL panel.”

Scale bars have now been inserted in this figure panel.

13. “Figure 2c: scale bars are missing from both lower panels.”

Scale bars have now been inserted in this figure panel.

14. “Figure 2g: low resolution makes it harder to judge the staining of cleaved caspase-3.”

(new Figure panel 2h) We have now incorporated higher resolution and higher magnification images to clearly demonstrate that skin from newborn Δ^{Ker} OTULIN shows a marked increase in the number of cleaved caspase-3-positive apoptotic cells relative to control skin.

15. “Supplementary figure 2: in panel (a) the authors show the wound size measurements up until d12 pw, however in panel (c) they show H&E stains at d14 pw that shows complete wound healing in the Δ KerOTULIN mice, notwithstanding the formation of the tumour-like lesion. Thus, it would be useful if the authors could show the wound size measurements up until d14 pw in panel (a).”

All wounds are completely closed at day 14 post-wounding. We included these data in a new Supplementary figure 2c.

16. “Figure 3a: please provide epidermal thickness measurements for the comparison between the different genotypes. Also, the H&E representative images are of different resolutions (as indicated by different scale bars), consider showing consistent images.”

We measured epidermal thickness for the different mouse lines and included these in new Figure panels 3c, 5f and 7d. We now also show representative images with similar scale in Figure 3a.

17. “Line 269: “IFN-response genes (IRGs)” are more commonly known in the literature as Interferon-stimulated genes (ISGs).”

We have changed the text, as suggested by the Reviewer.

18. “Figure 5b: Why the choice to look at gene expression in epidermal tail lysates instead of back skin non-lesion and lesion from Δ KerOTULIN, which would be a better comparison to the scRNA-seq data?”

As mentioned above, we used tail skin lysates because of the greater fraction of keratinocytes in these lysates compared to back skin lysates.

19. “Supplementary 2c and 3c: missing scale bars in some panels.”

Scale bars have been inserted.

20. “Figure 5c: The H&E panels are missing annotation and are of different resolution. Also, the third H&E panel depicts what looks like inflamed skin in the Δ KerOTULIN x IFNAR1^{-/-} mice therefore please provide epidermal thickness measurements across genotypes for comparison.”

We have modified the figure to make this more clear. We also clarified in the text that, while some Δ^{Ker} OTULIN-IFNAR1^{-/-} mice are completely protected from dermatitis, others still develop skin inflammation over time (Figure 5c-d).

We measured epidermal thickness for the different mouse lines and included these in new Figure panels and 3c, 5f and 7d. Here (Figure panel 5f), we also compared epidermal thickness in Δ^{Ker} OTULIN-IFNAR1^{-/-} mice that have or don't have skin lesions.

21. “Figure 7c: right panel in the H&E staining is missing scale bar.”
We inserted scale bars in all panels.

22. “Please provide clone and catalogue number details for all antibodies used in the manuscript.”
We included this information in the materials and methods section of the manuscript.

Reviewer #2:

“In this manuscript, authors report that OTULIN is a crucial regulator for maintaining skin cell homeostasis and preventing keratinocyte death and subsequent skin inflammation. OTULIN deletion in keratinocytes results in enhanced keratinocyte proliferation and cell death. TNFR1 deficiency or knockin expression of kinase-inactive RIPK1 or FADD and MLKL deletion prevents dermatitis development in the keratinocyte-specific OTULIN-deficient mice. In addition, the authors show that type 1 IFNs and IL-1b contribute to the skin inflammation in keratinocyte-specific OTULIN-deficient mice. These findings are interesting and provide new insight into this field; however, the current version of the manuscript has several issues that need to be addressed to strengthen the conclusions.”

Major comments:

1. “Fig. 1i shows increased NF-kB activity in the epidermal tail lysates of OTULIN deficient mice. However, OTULIN deficiency has no effect on TNF-induced NF-kB signaling in primary keratinocytes (Fig. 3d). The authors should discuss the potential mechanism by which OTULIN regulates keratinocyte signaling in vivo. Does OTULIN deficiency in primary keratinocytes promote induction of NF-kB and MAPK signaling by proinflammatory cytokines, such as IL-1b?”

We included new data demonstrating that Δ^{Ker} OTULIN keratinocytes are sensitized to cell death. Indeed, although OTULIN-deficient PMKs do not die when stimulated with TNF, they are sensitized to TNF-induced cell death when primed with IFN- γ prior to TNF-stimulation. We included these new findings in Figure 3 of the revised manuscript (Fig. 3d and e). In addition, we provide evidence that MCP-1 acts as a crucial chemokine attracting immune cells to the skin. We show that MCP-1 levels are upregulated in the skin of Δ^{Ker} OTULIN mice and demonstrate that dermatitis in Δ^{Ker} OTULIN mice is suppressed when mice are repetitively injected with neutralizing MCP-1 antibodies. These data are now shown in a new Figure panel 1f, in Figure 6f-g and Supplementary Figure 6.

2. “Lesional skin of Δ^{Ker} OTULIN mice have enhanced cell proliferation and cell death (Fig. 2). However, it is unclear whether the cells positive for cleaved caspase 3 and Ki67 are keratinocytes or infiltrating immune cells. Also, the authors should discuss how OTULIN deficiency increases keratinocyte proliferation.”

Double staining for keratin-14 (keratinocyte marker) and cleaved caspase-3 or Ki67 clearly demonstrates that cells positive for cleaved caspase-3 or Ki67 are indeed the keratinocytes. These immunostainings are now shown in Supplementary Figure 2a and b.

Our data demonstrating enhanced keratinocyte cell death and proliferation in Δ^{Ker} OTULIN mice suggest that the dermatitis in Δ^{Ker} OTULIN mice develops as a result of continuous keratinocyte apoptosis/necroptosis and compensatory keratinocyte proliferation, as was also shown in other tissues and models. Since inhibition of cell death (in TNFR1 deficient, RIPK1-D138N transgenic or FADD/MLKL deficient conditions) prevents dermatitis

development in Δ^{Ker} OTULIN mice, these data demonstrate that keratinocyte death drives proliferation as a compensatory mechanism. We shortly discuss this in the revised manuscript.

3. “Regarding Fig.2, what is the phenotype of keratinocyte proliferation and epidermal thickness in Δ^{Ker} OTULIN-TNFR1^{-/-}, Δ^{Ker} OTULIN-RIPK1D138N/D138N , Δ^{Ker} OTULIN/FADD/MLKL mice? Also, mechanistically, what is the relationship between aberrant cell death and cell proliferation in Δ^{Ker} OTULIN mice?”

We have now measured epidermal thickness for the different mouse lines and included these in new Figure panels and 3c, 5f and 7d.

Our data demonstrating enhanced keratinocyte cell death and proliferation in Δ^{Ker} OTULIN mice suggest that the dermatitis in Δ^{Ker} OTULIN mice develops as a result of continuous keratinocyte apoptosis/necroptosis and compensatory keratinocyte proliferation.

4. “It is interesting that OTULIN deficiency has no effect on TNF-induced cell death in primary keratinocytes (Fig. 3c). However, the data are not quite convincing, because TNF did not induce cell death in WT cells (untreated and TNF-treated cells had similar level of cell viability). Since TNFR1 deficiency rescues Δ^{Ker} OTULIN mice from dermatitis development, it is important to repeat this experiment using a higher dose of TNF α . If OTULIN deficiency indeed has no effect on TNF-induced cell death in primary keratinocytes, the authors should discuss how OTULIN may regulate cell death *in vivo*. Which cell death trigger could be regulated OTULIN?”

As mentioned above, we now included new data demonstrating that OTULIN-deficient keratinocytes are sensitized to TNF-induced cell death when primed with IFN- γ prior to TNF stimulation. These *in vitro* data are in line with our *in vivo* observations demonstrating cell death as a driver of skin pathology in Δ^{Ker} OTULIN mice.

5. “Fig. 3e shows a substantial reduction in the level of RIPK1 in Δ^{Ker} OTULIN keratinocytes. Is this result reproducible? Does OTULIN regulate RIPK1 stability?”

As shown in the figure below (Figure R4, for reviewers only), immunoblotting for RIPK1 in epidermal tail lysates could not confirm a difference in RIPK1 levels between OTULIN deficient and sufficient lysates, indicating that OTULIN does not regulate RIPK1 stability.

Figure R4 : Western blotting on epidermal tail lysates of 7-week old OTULIN^{fl/fl} (WT) and Δ^{Ker} OTULIN (KO) mice using antibodies detecting OTULIN and RIPK1. Anti-actin is shown as loading control.

6. “The finding that OTULIN regulates stem cell populations is interesting (Fig. 4f). Could the authors propose the possible signaling mechanism (which receptor pathway might be regulated by OTULIN)?”

Our observations that OTULIN deficiency affects stem cell populations suggests that OTULIN may regulate Wnt signaling, in line with previous studies (Rivkin et al., 2013; Wang et al., 2020). Wnt receptors, that also function as hair follicle stem cell markers such as Lgr5 or Lgr6, are involved in stem cell plasticity. However, more research is needed to

clarify OTULIN's role in Wnt signaling and stem cell regulation, which is subject of ongoing research and does not represent the focus of this study.

7. "What is the potential mechanism by which OTULIN-deficiency in keratinocytes induces innate immune cells infiltration (Fig. 4)?"

We provide new data showing that MCP-1 (a crucial chemokine in attracting immune cells to the skin) levels are upregulated in the skin of Δ^{Ker} OTULIN mice. Moreover, we now also show that dermatitis in Δ^{Ker} OTULIN mice is suppressed in conditions where MCP-1 is blocked using MCP-1 neutralizing antibodies. These data are shown in a new Figure panel 1f, in Figure 6f-g and Supplementary Figure 6.

8. "Fig.5: how does OTULIN regulate IFN and IFN-response genes? What's the phosphorylation level of TBK-1 and IRF3?"

IFNAR1 deficiency significantly protects Δ^{Ker} OTULIN mice from developing dermatitis, suggesting that OTULIN may regulate the pathways that are responsible for the production of IFNs, either indirectly by preventing overall inflammation or by direct control of IFN production. We have analysed p-TBK-1 and p-IRF-3 levels (as well as the level of unphosphorylated proteins) by Western blotting on lysates of LPS-stimulated PMKs. However, we could not observe major changes in signaling between the two genotypes (Figure R5, for reviewers only). Further studies are required to identify the pathway(s) regulated by OTULIN driving type I IFN production.

Figure R5 : Western blotting on lysates of PMK cultures from OTULIN^{fl/fl} (WT) and Δ^{Ker} OTULIN (KO) mice using antibodies detecting phosphorylated and unphosphorylated TBK1 and IRF3. Anti-actin is shown as loading control.

REVIEWERS' COMMENTS

Reviewer #1 (Remarks to the Author):

In the revised manuscript by Hoste, Lecomte et al., the authors addressed the majority of the raised points and have included considerable amount of new experimental evidence. All in all, the revised manuscript is very well-fashioned and nicely outlines the requirement of OTULIN in the homeostasis of keratinocyte and skin barrier integrity.

However, there are a couple of specific points that I would like the authors to address:

1. Fig 3f: the authors state that the residual OTULIN bands seen the figure is due to the feeder cells in the PMK cultures. This raises questions about the interpretation of the experiment. Is it plausible that the seemingly unperturbed NF- κ B and MAPK signalling in response to TNF is due to the response by the feeder cells? In relation to this, the authors show convincingly in the new Fig 7e that OTULIN deficiency results in loss of HOIP and SHARPIN in tail lysates whereas figure 3f shows residual amount of SHARPIN and HOIP that possibly originates from the feeder cells. The presence of the feeder cells in the culture thus seems to dilute any differences between the genotypes, which impacts on the conclusions from the experiment. For example: Fig 1h: increased Nfkb signaling in tail lysates (reduced IkBa and increased p-IkBa) vs Fig 3f: no change at homeostasis or following TNF in PMK cultures. Also, the clear deregulation of MCP-1 mRNA levels shown in Fig 1f is much less evident in Suppl. Fig. 3d. TNF-induced expression of Tnf and Mcp-1 (Suppl. Fig 3d) is stated to be unaltered in OTULIN-deficient PMKs but it appears there is a trend towards higher expression in KO cells relative to WT – would the difference perhaps be clearer without the contribution of mRNA from the feeder cells? Because of issues with the interpretation of the PMK culture experiments, this reviewer would advice to tone down the conclusions and/or include a note regarding the feeder cell issue.

2. In the unchanged supplementary Fig 4b, the representative flow cytometry plots show that all isolated T cells (CD3+) cells in the skin preps are also positive for TCR $\gamma\delta$ +, however, in Fig 4f there is also quantification of TCR α/β + and Tregs cells. This is confusing. It is also perplexing that in supplementary Fig 4f the abundance of Tregs is higher than of total $\alpha\beta$ T cells since Tregs are $\alpha\beta$ -positive.

3. For supplementary Fig 4c and 4e, some information on the markers (genes) that segregate the clusters should be included.

4. Figure 1h: If the distinct pattern of the Ikba and p-Ikba signal is a result of reblotting for total Ikba following p-Ikba, it is advised to clearly state this in the figure or legend to avoid any confusion about the reason for the similar pattern.

Specific points:

Line 215: Mistake in reference to Supplementary figure. Should be Suppl. Fig 3d

Line 221: Mistake in reference to Supplementary figure. Should be Suppl. Fig 3e

Reviewer #2 (Remarks to the Author):

The authors have adequately addressed my concerns, and the revised manuscript has been substantially improved.

Point-by-point response to reviewers

Reviewer #1 (Remarks to the Author):

1. Fig 3f: the authors state that the residual OTULIN bands seen the figure is due to the feeder cells in the PMK cultures. This raises questions about the interpretation of the experiment. Is it plausible that the seemingly unperturbed NF-kB and MAPK signalling in response to TNF is due to the response by the feeder cells? In relation to this, the authors show convincingly in the new Fig 7e that OTULIN deficiency results in loss of HOIP and SHARPIN in tail lysates whereas figure 3f shows residual amount of SHARPIN and HOIP that possibly originates from the feeder cells. The presence of the feeder cells in the culture thus seems to dilute any differences between the genotypes, which impacts on the conclusions from the experiment. For example: Fig 1h: increased Nfkb signaling in tail lysates (reduced IkbA and increased p-IkBa) vs Fig 3f: no change at homeostasis or following TNF in PMK cultures. Also, the clear deregulation of MCP-1 mRNA levels shown in Fig 1f is much less evident in Suppl. Fig. 3d. TNF-induced expression of Tnf and Mcp-1 (Suppl. Fig 3d) is stated to be unaltered in OTULIN-deficient PMKs but it appears there is a trend towards higher expression in KO cells relative to WT – would the difference perhaps be clearer without the contribution of mRNA from the feeder cells? Because of issues with the interpretation of the PMK culture experiments, this reviewer would advice to tone down the conclusions and/or include a note regarding the feeder cell issue.

We are confident that the conclusions we draw from the differences in protein levels as observed by Western blotting hold true. In Figure 1, epidermal tail lysates are shown that demonstrated enhanced NF-kB signalling. *In vitro*, we did not observe this difference, but this is not an effect from possible feeder contamination, as we repeated this experiment multiple times and always could see similar levels of NF-kB signalling in OTULIN-proficient versus -deficient PMKs, while the level of feeder contribution to these cultures differed. Also, in figure 1f *ex vivo* samples were analysed, while in Suppl. Figure 3d, *in vitro* samples were investigated. Again showing that many of the differences we see *in vivo* are not present in *in vitro* PMKs.

However, we did include a statement on the possible feeder contamination in PMK cultures: *'It should be noted that the residual OTULIN band observed in Δ^{Ker} OTULIN PMKs might originate from feeder cells that can still be present in PMK cultures.'*

2. In the unchanged supplementary Fig 4b, the representative flow cytometry plots show that all isolated T cells (CD3+) cells in the skin preps are also positive for TCR $\gamma\delta$ +, however, in Fig 4f there is also quantification of TCR α/β + and Tregs cells. This is confusing. It is also perplexing that in supplementary Fig 4f the abundance of Tregs is higher than of total $\alpha\beta$ T cells since Tregs are $\alpha\beta$ -positive.

We would like to point out that in Figure 4b the gating strategy for $\gamma\delta$ T-cells is shown, where we indeed gated for CD3+ $\gamma\delta$ + cells to discriminate $\gamma\delta$ T-cells. Within the T cell population a clear population of $\alpha\beta$ + cells is present, which is quantified in Suppl. Fig4f.

This reviewer is correct in pointing out the discrepancy in $\alpha\beta$ T-cell numbers versus Tregs. This is due to the fact that we permeabilized the cells for intracellular FoxP3 staining and hence could not gate for live cells, as we did for the other T-cell populations. We have now changed this by showing the numbers of Tregs as a percentage of CD45+ cells.

3. For supplementary Fig 4c and 4e, some information on the markers (genes) that segregate the clusters should be included.

We now included additional information on differentially expressed genes that were present in different subclusters of T-cells in the figure legend of Supplementary Figure 4, where possible.

4. Figure 1h: If the distinct pattern of the I κ B α and p-I κ B α signal is a result of reblotting for total I κ B α following p-I κ B α , it is advised to clearly state this in the figure or legend to avoid any confusion about the reason for the similar pattern.

We included additional information in the figure legend stating that immunoblotting was first performed for phospho-I κ B α and consecutively for I κ B α on the same blot.

Specific points:

Line 215: Mistake in reference to Supplementary figure. Should be Suppl. Fig 3d

Line 221: Mistake in reference to Supplementary figure. Should be Suppl. Fig 3e

Supplementary figure 3 has been edited and new panels have been inserted. All references to this figure are now correct.

Reviewer #2 (Remarks to the Author):

The authors have adequately addressed my concerns, and the revised manuscript has been substantially improved.

We thank this Reviewer for taking the time to assess the revised version of our manuscript.